# Drug target prediction through deep learning functional representation of gene signatures

Hao Chen [1,2,3] ✉, Frederick J. King [1], Bin Zhou[1], Yu Wang[1], Carter J. Canedy[1], Joel Hayashi[1], Yang Zhong[1], Max W. Chang [4], Lars Pache [5], Julian L. Wong[1], Yong Jia[1], John Joslin[1], Tao Jiang [2], Christopher Benner [4], Sumit K. Chanda [6] & Yingyao Zhou [1] ✉

Many machine learning applications in bioinformatics currently rely on matching gene identities when analyzing input gene signatures and fail to take advantage of preexisting knowledge about gene functions. To further enable comparative analysis of OMICS datasets, including target deconvolution and mechanism of action studies, we develop an approach that represents gene signatures projected onto their biological functions, instead of their identities, similar to how the word2vec technique works in natural language processing. We develop the Functional Representation of Gene Signatures (FRoGS) approach by training a deep learning model and demonstrate that its application to the Broad Institute's L1000 datasets results in more effective compound-target predictions than models based on gene identities alone. By integrating additional pharmacological activity data sources, FRoGS significantly increases the number of high-quality compound-target predictions relative to existing approaches, many of which are supported by in silico and/ or experimental evidence. These results underscore the general utility of FRoGS in machine learning-based bioinformatics applications. Prediction networks pre-equipped with the knowledge of gene functions may help uncover new relationships among gene signatures acquired by large-scale OMICs studies on compounds, cell types, disease models, and patient cohorts.

Large-scale OMICs investigations of biological systems can produce "gene signatures" represented as lists of gene candidates that are surmised to be relevant to the biological activity under interrogation. Similarity comparisons between gene signatures derived from related biological assays or technology platforms are used to predict functional relationships between compounds, genes, and proteins. For example, the Library of Integrated Network-Based Cellular Signatures (LINCS) L1000 program systematically generated 1.3 million gene expression profiles in human cell lines with ~22,000 genomic and 20,000 pharmacological perturbations[1]. As compounds and short hairpin RNAs (shRNA)/complementary DNAs (cDNA) perturbing the same target are expected to generate correlated modulations in downstream gene expression, similar L1000 transcriptional signatures can offer an unbiased data-driven mechanism to identify compound-target pairs[2-6].

[1]Novartis Biomedical Research, 10675 John Jay Hopkins Drive, San Diego, CA 92121, USA. [2]Department of Computer Science and Engineering, University of California, Riverside, 900 University Avenue, Riverside, CA 92521, USA. [3]Computational Biology Department, School of Computer Science, Carnegie Mellon University, Pittsburgh, PA 15213, USA. [4]Department of Medicine, University of California, San Diego, 9500 Gilman Drive, La Jolla, CA 92093, USA. [5]NCI Designated Cancer Center, Sanford Burnham Prebys Medical Discovery Institute, La Jolla, CA 92037, USA. [6]Department of Immunology and Microbiology, Scripps Research, La Jolla, CA 92037, USA. ✉e-mail: hchen4@andrew.cmu.edu; yingyao.zhou@novartis.com

The underlying molecular governance for a single gene signature can be extracted using well-established statistical frameworks known as pathway enrichment analyses, where the calculations are based on the number of gene members that overlap with known pathways curated by bioinformatics knowledgebases[7,8]. Unfortunately, adopting similar statistical metrics to compare two experimentally derived gene signatures are ineffective in many cases: influenza host dependency factors identified by eight published studies only shared a modest overlap[9]; very few genetic interactions are replicated across multiple studies measuring synthetic lethality[10]; and cell type-specific gene signatures identified from two mouse embryonic single cell studies showed little overlap[11]. According to studies comparing the performance of compound target predictions using multiple data types[12–14], predictions based on transcriptional response alone tended to underperform when compared to models using structural or pharmacological features[12]. This challenge occurred regardless of whether the L1000 landmark genes or their extrapolated whole-transcription counterparts are used. Therefore, understanding and addressing the limitations in existing gene signature comparison algorithms can enhance our understanding of how transcriptional data can be employed to predict the mechanism of action of small molecules. As target identification of these chemical leads in cell-based assays presents a major impediment to their progression in drug discovery, effective elucidation of their efficacy and potential side effects will accelerate the drug discovery and development process[15].

We hypothesized that the difficulty of comparing experimentally derived signatures originates from the fact that each signature consists of only a sparse sampling of the genes underlying regulated pathways. For example, if we randomly sample 10 genes from a hypothetical 100-gene pathway twice, the chance of having three or more common genes is only 6%, despite representing the same pathway. Such sparseness is intrinsic to all experimental signatures, as it can arise from the technical alterations of signal in RNA-seq studies[16], read dropouts with lower gene expression levels[17], the regulatory variations in transcriptional factor binding sites[18], the stochastic gene expression in single cell studies[19], or the rare variants in genome-wide association studies[20]. In most popular gene-signature-similarity calculations, genes are treated as identifiers and their underlying functional roles are ignored. For example, the Connectivity Map (CMap) score used in the LINCS workflow evaluates similarities based on the weighted Kolmogorov-Smirnov enrichment statistic[21]. Likewise, BANDIT[12] uses the Pearson correlation to measure the degree of similarity for two gene signatures. These methods, along with other recent machine-learning-based models[5,21], compute gene signature similarities purely by matching gene identities (Fig. 1a), e.g., *TLR7* and *MYD88* are treated independently even though they play remarkably similar biological roles in innate immune signaling.

The weakness in extracting functional relationships from gene signatures by gene identity counting has a strong analogy in the natural language processing (NLP) field. Early NLP analyses used one-hot encoding of words: each word was encoded by its identity and two words such as "cat" and "kitty" were considered as equally distant as "cat" and "rock" (Fig. 1a). This limitation has been addressed by exciting breakthroughs in various NLP machine-learning applications following the introduction of word-embedding technologies, such as word2vec[22]. The semantic meanings of words can now be accounted for in word vectors leading to "cat" and "kitty" being correctly recognized as synonyms.

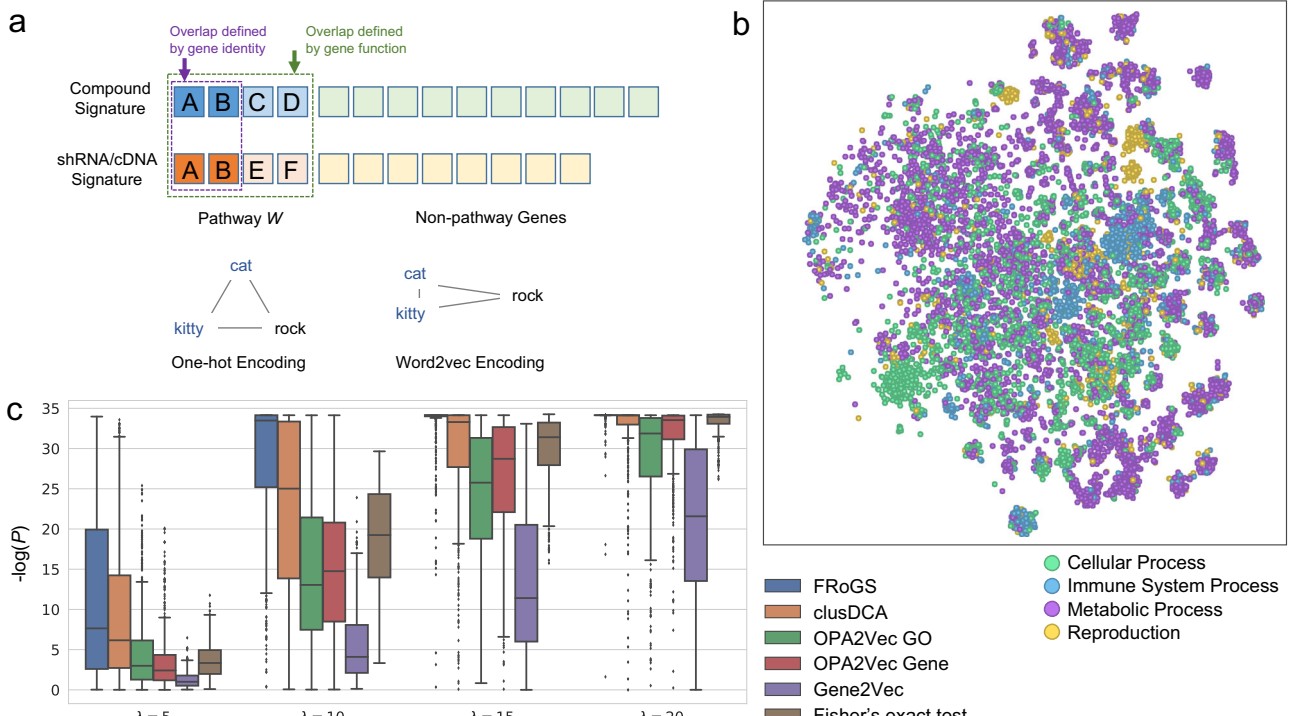

**Fig. 1 | FRoGS can extract weak pathway signals. a** Comparison between two hypothetical gene signatures. Only gene A and B in Pathway *W* are considered overlapped based on gene identity (top), similar to the use of one-hot encoding in NLP. Genes A-F contribute to signature overlap if all genes of the same functions *W* are considered (bottom), similar to the use of word2vec. **b** t-SNE projection of gene embedding vectors, where each marker represents a gene. Markers are colored by their top-level functions annotated in GO. **c** Each of the 460 Reactome[41] pathways was used to simulate foreground gene signatures generated under varying signal levels with $\lambda$ at 5, 10, 15, and 20. The separation between foreground-foreground and foreground-background pairs is defined as $-\log_{10}(p)$ based on the one-sided Wilcoxon signed-rank test ($n = 200$ simulations). The larger the value, the more sensitive the method can separate the two types of signature pairs. Each pathway contributes to one data point in each box plot. Box-and-whisker plots show the median (center line), 25th, and 75th percentile (lower and upper boundary), with $1.5 \times$ inter-quartile range indicated by whiskers and outliers shown as individual data points. Source data are provided as a Source Data file.

Inspired by how NLP captures word semantics, we hypothesized that capturing functional instead of identity overlap between two gene signatures could overcome the sparseness limitation of identity overlap and lead to greater sensitivity in extracting common pathways induced by co-targeting compound-shRNA/cDNA pairs (Fig. 1a). Although methods were previously proposed for the representation learning of individual genes[23–28] and some of them were applied to drug target predictions[3,26,29,30], they either do not directly encode genes' functions or do not embed signatures involving multiple functionally related genes. A critical advance of this study is to introduce a form of "word2vec" for bioinformatics named Functional Representation of Gene Signature (FRoGS), where FRoGS vectors encode known human genes' functions based on hypergraphs formed by Gene Ontology (GO)[31], as well as their empirical functions proxied by experimental expression profiles from ARCHS4[32]. The machine-learning utility of FRoGS is demonstrated in the L1000-based compound target prediction application introduced above. Different from previous protein-network-based methods[4,33], we used a neural network that takes the FRoGS vector representations as input to directly compute the similarity between the signatures of compound perturbation and target gene modulation. Compared to multiple L1000-based transcriptional models, our FRoGS-based model significantly outperformed identity-based methods and methods based on other gene-embedding schemes. When existing models based on other data sources were augmented with our model, consistent performance boosts were observed, suggesting the broad utility of FRoGS to enhance comparative analysis of gene-signatures in large-scale datasets[34–38].

In this study, we present the FRoGS vector, which is a functional embedding of human genes. We showcase its potential with an example machine learning application, where FRoGS substantially improves the success rate of discovering compound targets.

## Results

### FRoGS extracts weak pathway signals from gene signatures

FRoGS vectors were trained such that individual human genes are mapped into high dimensional coordinates encoding their functions. Our deep learning model aimed to assign coordinates so that neighboring genes tend to share similar GO annotations as well as correlated experimental expression profiles as defined in ARCHS4[32]. During gene set analysis, vectors associated with individual gene members are then aggregated into a single vector encoding the whole gene set signature (see Methods).

For the purpose of visually confirming the validity of the FRoGS vectors, we used a 2-dimensional t-SNE projection[39] (Fig. 1b) to confirm whether individual genes were grouped based on their functions in the embedding space, in a manner similar to how synonyms are co-located in the word2vec embedding. Genes closely positioned in the same clusters tend to share the same biological function (i.e., node color in Fig. 1b) with $p < 10^{-100}$, indicating that similarities among FRoGS embeddings indeed proximate their functional closeness. To further validate our hypothesis that functional gene-set embedding could boost sensitivity in detecting shared functionality between two perturbation signatures, we simulated experimentally derived signature pairs. Specifically, we randomly generated two foreground gene sets and a background gene set, each with 100 genes, for a given pathway $W$. The foreground gene sets simulated experimentally derived gene signatures from two perturbations co-targeting $W$. Both were seeded with $\lambda$ random genes within $W$ and $100 - \lambda$ random genes outside $W$, while the background gene set contains no gene in $W$. More sensitive methods were expected to find the proper foreground gene signature pair $(S_{fg}, S'_{fg})$ to be more similar than the foreground-background pair $(S_{fg}, S_{bg})$. Parameter $\lambda$, the number of pathway genes, modulates the strength of the pathway signals in foreground gene sets (see Methods). We compared the performance of multiple state-of-the-art gene and GO embedding methods, including OPA2Vec[24], Gene2vec[23],

clusDCA[25], and Fisher's exact test (Supplementary Note 1). Comparison with Fisher's exact test was particularly informative, as it represents the popular gene identity-based similarity measurement that is currently adopted by most bioinformatics algorithms, and is conceptually similar to the principles underlying the CMap score[1,40]. The above-described sampling process was repeated 200 times and the resulting similarity score distributions were compared using one-sided Wilcoxon signed-rank test to characterize if the $(S_{fg}, S'_{fg})$ similarity scores were larger than the $(S_{fg}, S_{bg})$ similarity scores, where higher -log($p$) values are more desirable. We considered all 460 human pathways with numbers of associated genes in the range of 50–200 as captured in the Reactome database[41] and summarized the results in Fig. 1c. We observed most embedding methods outperformed Fisher's exact test when challenged with weak signals ($\lambda = 5$) (Fig. 1c), while Fisher's exact test performed well only under strong signals ($\lambda \geq 15$). FRoGS remained superior across the whole range of $\lambda$ values. More detailed analyses using pathway R-HSA-5576891 (cardiac conduction) as an example are provided in Supplementary Fig. 1a, further supporting the conclusion presented in Fig. 1c, even as the size of gene lists vary (Supplementary Fig. 1b–c). These observations not only explain the difficulty encountered by current gene identity-based algorithms in extracting weak molecular signals, but also demonstrate that FRoGS provided a more sensitive approach to gene signature overlap analysis, providing the foundation for its application in bioinformatics gene signature comparisons.

### FRoGS recalls more known compound targets

With each compound and genomic perturbation represented by an aggregated FRoGS signature vector corresponding to their extrapolated whole-transcriptome L1000 profiles[1] (see Methods), we trained a Siamese neural network model that applies the same network to a pair of signature vector inputs representing the transcriptional landscape after compound perturbation or shRNA/cDNA modulation (knockdown or overexpression of a target gene, respectively) (Fig. 2a). Each compound signature $c$ was paired with every genomic signature $g$ acquired within the same cell line to predict the probability of a $(c, g)$ pair sharing the same target within the given biological context. Our positive training dataset consisted of 2340 $(c, g)$ pairs formed between 1438 compounds and 499 targets annotated in the L1000 database[42]; compounds with more than five targets were excluded from the training to reduce the impact of polypharmacology. Importantly, we adopted a balanced data sampling strategy for model training, in which each target occurred in equal frequency in both positive and negative training pairs, to avoid bias towards popular known targets (Supplementary Fig. 2 and Supplementary Note 2).

The multiple predictions obtained for the same $(c, g)$ pair across different cell lines and perturbagen types were further aggregated into a consensus target ranking, and then mapped into a probability score using an adjusted logistic regression (LR) model (see Methods), which is referred to as Model $L$ hereafter, where $L$ stands for L1000. A well-adopted convention for comparing transcriptional models is to calculate the recall[3,5], i.e., the percentage of compounds having their Broad-annotated known targets[42] predicted among the top $N$% of the candidate list, with $N$ set to 5 here (see Methods). For comparison, we trained the same Siamese networks using gene signatures represented by other state-of-the-art gene and GO embeddings as input. The recall-rank plot in Fig. 2b demonstrated a significant performance boost in our FRoGS-based model compared to other approaches. With the baseline recall value estimated to be $8.0 \pm 2.7$% based on 100 random permutations, CMap[1,40] and OPA2Vec GO[24] methods performed similarly to random models with recalls at 9.6% and 9.5%, respectively. Fisher's exact test, another identify-based model, performed poorly at 12.3%. The other three methods, OPA2Vec gene[24], Gene2Vec[23], and clusDCA[25], obtained recalls of 15.9%, 20.7%, and 24.9%, respectively. Our FRoGS embedding-empowered prediction model achieved a recall

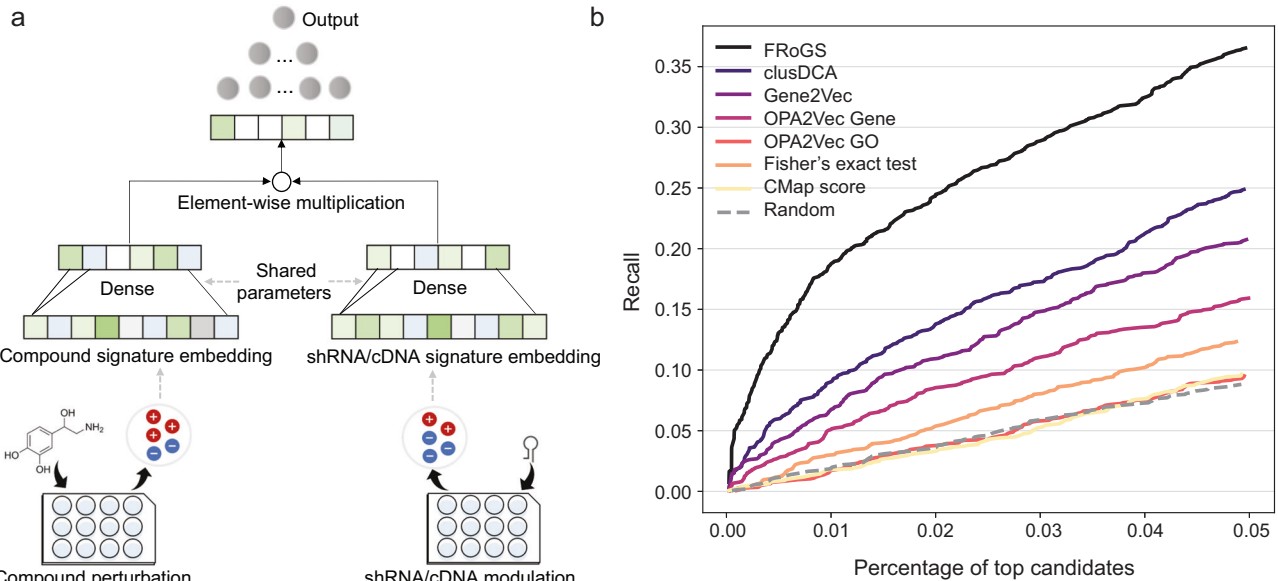

**Fig. 2 | FRoGS model predicts compound-target associations. a** The neural network architecture predicting the probability of a compound binding to a target based on their L1000 gene set signature embeddings. **b** The comparison of multiple L1000-based prediction models. The FRoGS model performed the best and CMap score, OPA2Vec GO performed similarly to random models. Source data are provided as a Source Data file.

of 36.3%, marking a significant advancement in the application of L1000-based gene signatures for compound target prediction. We also compared our FRoGS model with additional state-of-the-art L1000-based compound target prediction models[3-5] and demonstrated our model could achieve higher top-30 and top-100 accuracy scores[4] (Supplementary Fig. 3, Supplementary Note 2).

## FRoGS predicts compound targets supported by structure and activity data sources

The application of Model $L$ to all compound-gene pairs, captured in the full L1000 dataset regardless of whether target annotations were available, resulted in the prediction of 780,438 compound-target pairs with probability values above 0.8. The predictions on the training dataset were carried out by fivefold cross validation without data leak. Predicted compound-target pairs found in the Broad annotation were flagged as "known"[42]. We inspected relevant Novartis historical $pIC_{50}$ data for experimental support of a target prediction; those pairs with $pIC_{50}$ (measure of compound potency) $\leq 1\,\mu M$ against the relevant target in enzymatic or cellular assays are flagged as "$pIC_{50}$". Queried compounds with a structure–activity relationship (SAR) probability above 0.95 to a reference compound of the same predicted target are flagged as "structure sure", and those with SAR probability between 0.8 and 0.95 are flagged as "structure likely" (Supplementary Note 3).

A compound's pharmacological activities are useful for target prediction in the knowledge-based framework[13]. They rely on the guilt-by-association (GBA) principle that compound pairs sharing similar activity patterns across a large assay panel tend to interact with the same targets or perturb the same underlying pathway[43]. We therefore trained several activity-based models using compound activities for validation purposes in biological assays, including a profile-quantitative structure-activity relationship (pQSAR) dataset[44], Promotor Signature Profiling (PSP) dataset[45], and $NCI_{60}$ growth inhibition dataset[46], covering 1601, 93, and 321 Broad compounds, respectively (see Methods, Supplementary Note 4).

Due to their orthogonal nature (Supplementary Note 5, Supplementary Fig. 4), predictions from structural and activity models can serve as in silico validations for Model $L$ predictions. Predictions flagged as "known", "$pIC_{50}$", or "structure sure" were considered as "super-confident" predictions that were either validated or expected to be

validated. Predictions labeled "structure likely" were the "high-confident" SAR extrapolations of existing target annotations. Candidates predictable by one of the activity models were considered "medium-confident". In total, 2491 predictions fall into the three categories assigned based on their best supporting evidence (Fig. 3a). Below we enumerate a few "known" cases that otherwise would have been placed into different categories, as they provide an initial validation of successful target deconvolution for orphan compounds solely based on transcriptional signatures.

17 (0.7%) predictions are "structure sure". Examples include afimoxifene (BRD-K93754473), which was correctly predicted to target estrogen receptor 1 (ESR1) (score 0.91) and confirmed by its structural similarity with tamoxifen (BRD-K04210847) (SAR probability 0.97). Next, 88 predictions (3.5%) are considered "structure likely". For example, nitrendipine (BRD-A02006392) was predicted to block cholinergic receptor muscarinic 3 (CHRM3) (score 0.95), where its SAR probability against the reference compound, nicardipine (Fig. 3b), was 0.85. A total of 383 predictions (15%) with SAR probability above 0.8 endorse the assumption that Model $L$ can recover true compound-target associations as judged by SAR. The other 85% of the predictions cannot be inferred by SAR but are supported by other lines of evidence, which suggests the unique advantage of FRoGS-based transcriptional models in recalling novel targets without relying on chemical structure features.

For "medium-confident" predictions, 1756 (70%) were supported by the pQSAR activity matrix spanning 1003 unique gene targets. The prediction that perphenazine (BRD-K10995081) targets the dopamine receptor D1 (DRD1) (score 0.88, Fig. 3c) was supported by the high pQSAR model score (0.94) shared between perphenazine as the query and prochlorperazine (BRD-K19352500) as the reference. Similar high confidence values were obtained for mephentermine (BRD-K18194590) targeting histamine receptor H1(HRH1), supported by pQSAR (0.94) with respect to the reference compound pseudoephedrine (BRD-K91315211) (Fig. 3d). These functional similarities align with mephentermine's ability to indirectly induce noreprinehrine[47,48]. Similarly, 23 (0.9%) predictions are supported in the public $NCI_{60}$ dataset[46]. Examples include trametinib (BRD-K12343256), which was predicted to be a MEK1/2 allosteric inhibitor (score > 0.82), as it demonstrated similar biological activity profiles in reported $NCI_{60}$

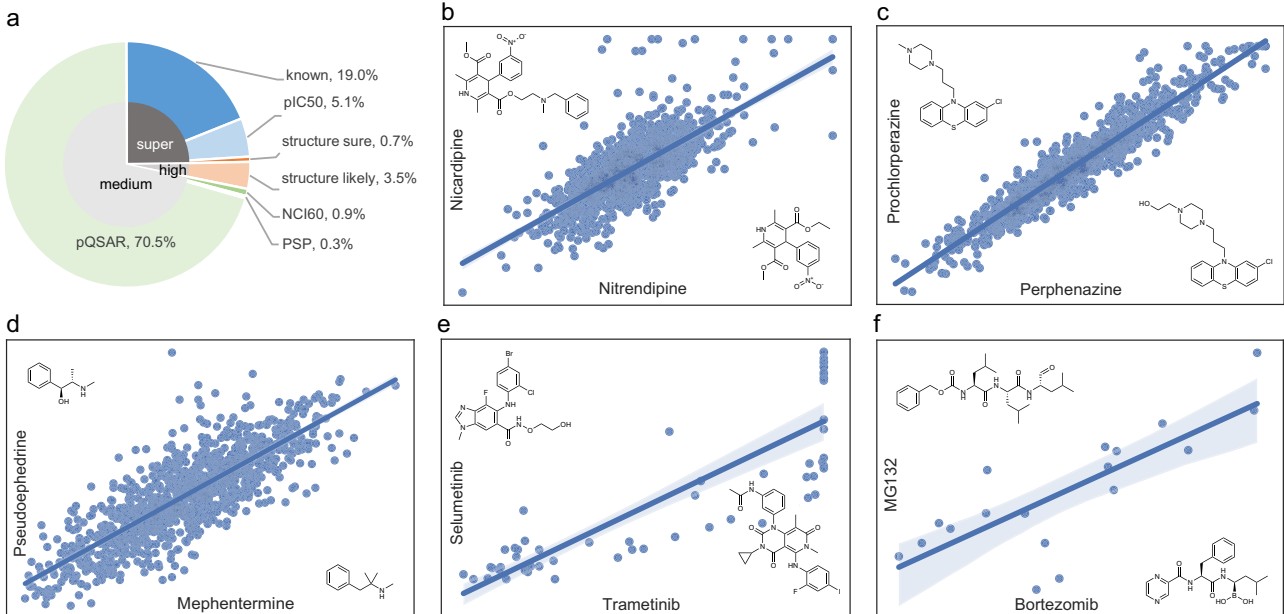

**Fig. 3 | Model _L_ predictions are supported by multiple orthogonal data sources.** **a** Validation categories for 2491 Model _L_ predictions. **b** A "structure-likely" example for nitrendipine targeting CHRM3 is supported by its structural similarity to the reference analog nicardipine. **c–d** Perphenazine targeting DRD1 and mephentermine targeting HRH1 are strongly supported by the pQSAR dataset. **e** Trametinib targeting MEK1/2 is supported by the NCI$_{60}$ dataset. **f** Bortezomib targeting BSMP1 is supported by the PSP dataset. Both axes in (**b–e**) are normalized Z-score assay activities and in (**f**) are GI$_{50}$ scores. Error bands in (**b–e**) represent the 95% confidence interval of the regression fit. Source data are provided as a Source Data file.

dataset ($r^2 = 0.78$) against selumetinib (BRD-K57080016) despite their low structural similarity (0.13) (Fig. 3e). Both molecules interact with the same allosteric pocket on MEK1/2 but contact different amino acid residues[49]. Lastly 8 (0.3%) predictions are supported in the PSP database[40] across a panel of 41 reporter gene assays. An example is bortezomib (BRD-K88510285) as the query and MG132 (BRD-K60230970) as the reference. These molecules share low structural similarity (0.6), yet the PSP correlation ($r^2 = 0.78$) suggests both target the proteasome 20S subunit beta 1 (PSMB1)[50] (Fig. 3f).

### FRoGS-based model augments compound activity models to predict a high-quality compound-target network

Model _L_ outperforms activity-based target prediction models based on NCI$_{60}$ (recall 17.2%) and PSP (recall 9.6%) datasets, marking a significant improvement compared to previous studies[12–14] (Supplementary Table 1). Comparing Model _L_ against the strongest model, pQSAR-activity-based target prediction, Metascape enrichment analyses[51] suggest that outside the broad overlap in predictions between two models, Model _L_ predicts distinct targets that are not recoverable by Model pQSAR and vice versa, indicating the unique strength of each model (Supplementary Fig. 5–6, Supplementary Note 6). As Model _L_ relies on transcriptional responses, while Model pQSAR is influenced by the enrichment of target classes covered by in-house assays and by focused compound screening libraries, the implicit bias of each approach to identify different targets is likely inherited from their data sources.

Given unique capabilities of Model _L_, we next demonstrated that our FRoGS-based Model _L_ can be combined with existing activity-based models to boost compound target prediction performance. For that, we integrated Model _L_ with the Model pQSAR into a multimodality LR predictor (Supplementary Note 7). The combined model significantly improved the performance of Model pQSAR from 0.70–0.74 ($p = 2 \times 10^{-4}$) (Supplementary Fig. 5a). Activity models based on NCI$_{60}$ and PSP datasets were also tested and showed consistent results (Supplementary Fig. 7, Supplementary Table 1). Multiple viable options of combining Model _L_ with the activity models were also evaluated as (Supplementary Note 8, Supplementary Fig. 8–9).

The combined model (Model _L_ with Model pQSAR) predicted a compound-target network comprising 1598 compounds, 682 genes, and 146,749 associations, where the probability was above 0.8 and the target was in the top 5% (Fig. 4a). We extracted subnetworks formed by compounds and their top candidate targets for each separate anatomical therapeutic chemical (ATC) classification according to DrugBank[52]. Ontology enrichment analyses[51] on the targets within each subnetwork showed clear biological agreements between the enriched pathways/ processes and the ATC labels. For example, targets predicted for the "other antineoplastic agents" network (ATC code L01X) (Fig. 4b) are highly enriched in "protein phosphorylation" processes (Fig. 4c) and depict associations between kinase targets and small molecules that are both well-established modulators of cancer biology. These include "rapalogues," which target mTOR, EGFR, and CDK4/6 inhibitors. These different mechanisms of action have related but distinct networks. Within this 129-edge subnetwork, 19% of edges are predictable by Model _L_ alone. 19% of the predicted edges are either "known" or "pIC$_{50}$" and 54% are supported by pQSAR evidence, implying high predictive quality.

Among all the network edges, 2183 (1.5%) are supported by all three models: Model _L_, Model pQSAR, and the combined model. Furthermore, 4566 (3.1%) edges found support in orthogonal data sources (not counting pQSAR), including 113 by NCI$_{60}$, 58 by PSP, 502 by structure similarity, 1271 by experimental pIC$_{50}$ data, and 2622 are "known" interactions. Combining these two subsets, we obtained a total of 6296 (4.5%) conceptually higher-quality predictions between 1406 compounds and 448 targets. This subset is released as a community resource that proposes valuable hypotheses for the target deconvolution of L1000 compounds, as well as supplies tool compounds for perturbing proteins of interest (Supplementary Data 1). Of note, 57% of the network edges from the combined model and 24% of the higher-quality subset would have been missed without the contribution of FRoGS-based Model _L_.

### Experimental validation of kinase inhibitors predicted by FRoGS

Kinases comprise 27% of compound targets in the Broad annotation, which is consistent with kinases accounting for over 30% of drug

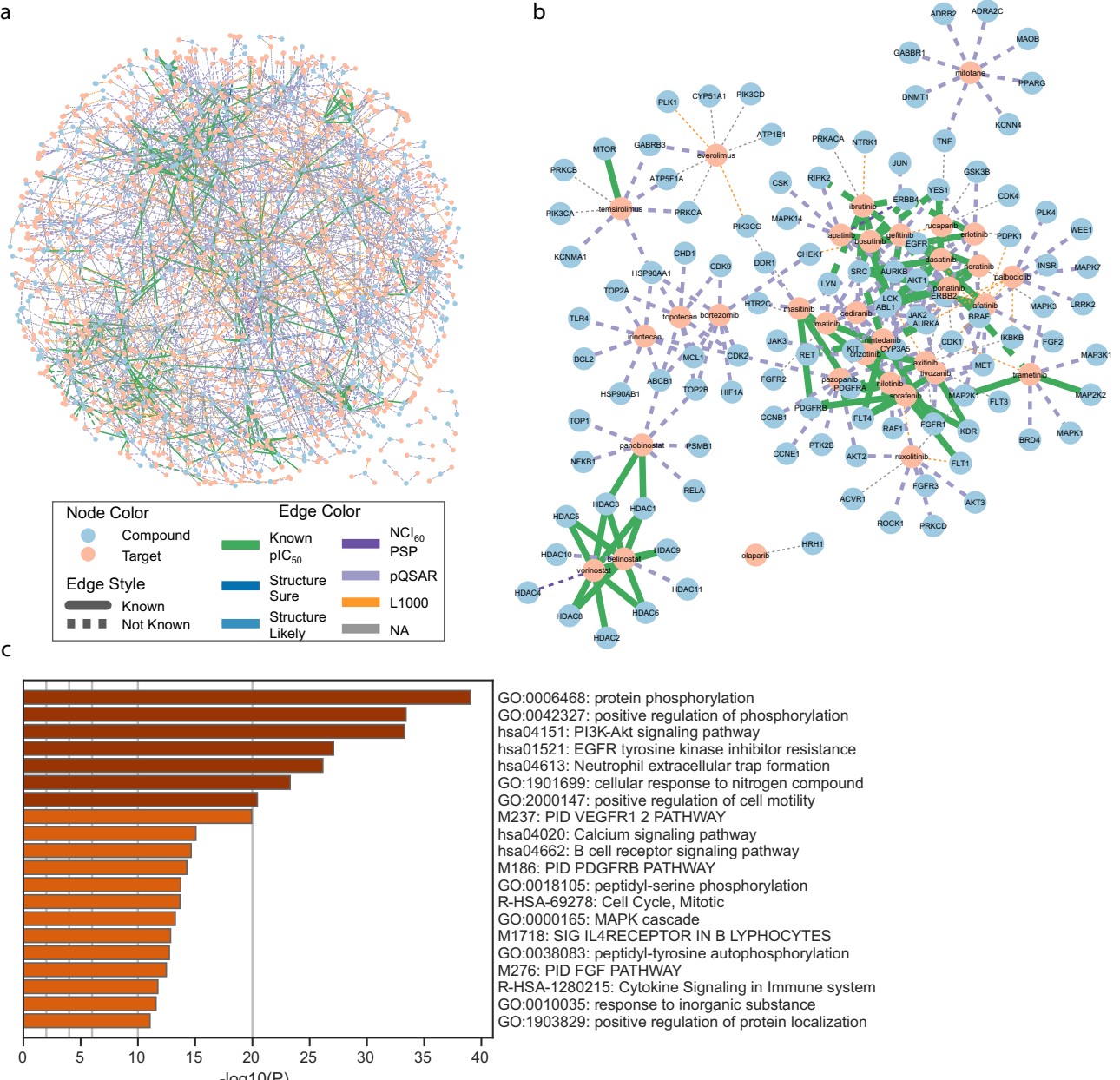

**Fig. 4 | The compound-target network predicted by the combined model. a** The network simplified by keeping ≤ 5 best-scoring targets per compound and ≤ 10 best-scoring compounds per target. Styles of nodes and edges are explained in the figure legend. **b** Network for L01X "other antineoplastic agents". **c** Targets in L01X subnetwork show the strong enrichment of pathways related to tyrosine phosphorylation and kinase-related signal transduction processes. One-sided fisher exact test, statistics including multi-test adjusted *p*-values are in source data. Source data are provided as a Source Data file.

discovery efforts worldwide[53]. Evaluation of predictions within our released network from molecules that interact with this target class is of primary importance. We selected all 1,116 compounds that were predicted to bind any of a set of 19 kinases and carried out experimental profiling to further validate these predictions (see Method). Figure 5a shows that in 18 out of 19 assays, predicted kinase binders have a significantly higher confirmation rate than those not predicted to bind (Supplementary Table 2). In the case of KDR, 70 out of 403 (17%) predictions were confirmed versus 9 out of 713 (1.9%) for negative predictions, signifying a 14-fold enrichment ($p = 1.2 \times 10^{-23}$). Primary hits active in no more than three assays were subjected to secondary dose-response profiling against nine kinases that had the largest number of target-specific primary hits, including PDGFRA, KIT, and EGFR. Dose-response profiling validated the single-dose profiling results (Fig. 5b, Supplementary Fig. 10, Supplementary Table 3).

Figure 5c shows the IC$_{50}$ binding heatmap of 191 compounds across the 9 assays, where 96 compounds bind to three or fewer kinases (Supplementary Data 2). Kinase targets clustered based on compound activity patterns generally reflect their functional lineage relationship, which implies the biochemical validity of the data. Among the 479 true positive (TP) predictions, 213 (44.5%) that were not predicted by Model pQSAR alone were rescued by Model *L* or the combined model, which demonstrates the important contributions of FRoGS-enabled Model *L*.

Commonly accepted target annotation for a compound is often incomplete. An example of this is A-443644 (BRD-K38615104), which is generally referred to as an AKT inhibitor. We predicted and demonstrated that this compound has activity against GSK3b, which also has been described by others[54]. Among its six predicted kinase targets, four were not predicted without the contribution of Model *L* (Supplementary Data 2). The approach also provided additional insights to

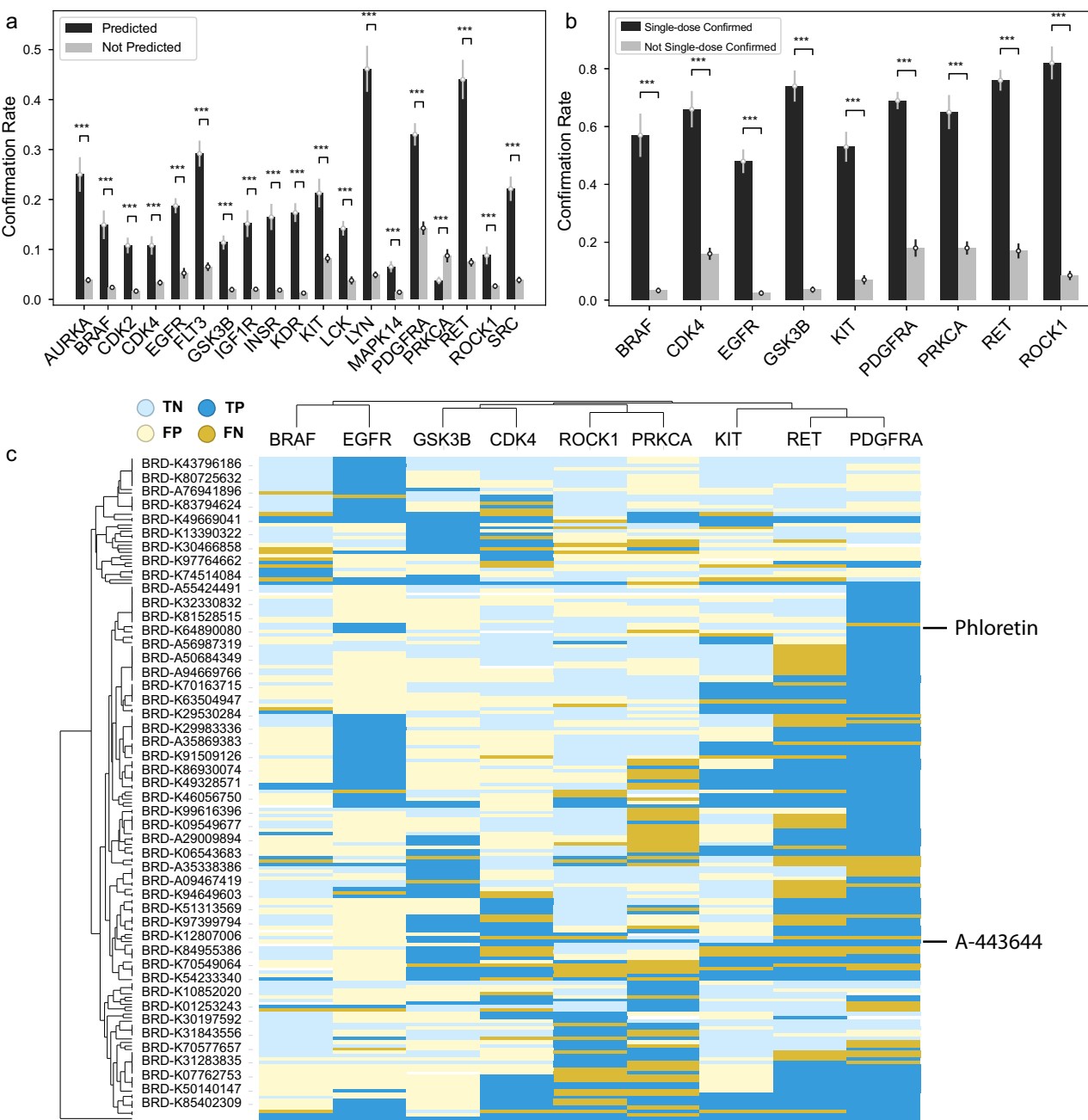

**Fig. 5 | Experimental confirmation of predicted kinase inhibitors. a** Single-dose confirmation rate for predicted kinase inhibitors (black) versus those not predicted to bind (gray). **b** Dose-response confirmation rate for compounds validated by single-dose confirmation versus those not validated. Error bars, mean ± SD, one-sided chi-square test. ***$p < 0.001$. **c** The $IC_{50}$ heatmap of 191 compounds across 9 kinases. True positives are in dark blue and false negatives are in dark yellow. Light blue indicates true negatives and light yellow is for false positives. Overall, 50% of compounds are selective (bound in ≤3 assays). Statistics are provided in Supplementary Table 2–3.

the mechanism of action of phloretin (BRD-K15563106), a compound with notable cell-based activity (e.g., differential cytotoxicity across the NCI60 panel) yet a poorly defined mechanism of action[55]. Phloretin is commonly referred to as an inhibitor of a sodium/glucose transporter, yet others have suggested it also may act as a kinase inhibitor[55,56]. The kinase screening data confirmed this latter activity against EGFR and other receptor tyrosine kinases, which provides insight into Phloretin's cytotoxicity.

### The discovery of ligands for aryl hydrocarbon receptor

We also investigated the use of the predictive model for non-kinase targets. The aryl hydrocarbon receptor (AhR) is implicated in many diseases that are driven by immune/inflammatory processes,

including major depressive disorder, multiple sclerosis, rheumatoid arthritis, asthma, and allergic responses[57]. AhR antagonists can expand human hematopoietic stem cells (HSC) ex vivo and can facilitate clinical HSC therapy[58]. Our model predicted 369 compounds potentially targeting AhR, of which 333 were available for profiling. AhR agonist, AhR antagonist, and CellTiter-Glo toxicity assays were utilized to screen these compounds. Initial testing at 50 μM, in quadruplicates, identified 128 compounds that are active in at least one of the AhR assays. Follow-up dose-response screens led to 76 compounds that were confirmed to be either AhR agonists or antagonists ($IC_{50} < 50$ μM) without apparent cytotoxicity (Fig. 6a, Supplementary Data 3), which corresponds to a 23% confirmation rate for our model predictions.

Among the confirmed binders, 3 known AhR agonists were recalled by our analysis, including ITE[59], diindolylmethane[60], and leflunomide[61] (Fig. 6b–d). Consistent with their observed antagonist activities, curcumin, resveratrol, and quercetin were previously shown to indirectly activate AhR by interfering with an endogenous AhR ligand FICZ[62] (Fig. 6e–g). Furthermore, publications report resveratrol as an antagonist[63], menadione to transcriptionally downregulate AhR in vivo[64] (Fig. 6h), taxifolin suppresses gastric cancer by regulating AhR/CYP1A1[65] (Fig. 6i), and isoliquiritigenin reduces the DNA-binding activity of AhR[66] (Fig. 6j).

In total, 38 compounds within this set were supported by Model pQSAR, while the other 38 (50%) compounds were predicted only with the contribution of Model $L$. This further underscores how Model $L$ can be effectively integrated with other OMICs data to improve compound target predictions. None of these compounds could have been discovered based on their chemical features as they are not structurally similar to any of the annotated AhR binders. AhR represents a difficult target class for prediction, as there are only 6 reference compounds among the 369 predictions (1.6%) and 45% of the predictions have no additional supporting evidence other than the combined model itself. The high confirmation rate implies our compound-target network is expected to contain rich biochemical associations that can be an invaluable resource for the biomedical research community to bootstrap drug discovery projects.

## Discussion

A significant contribution of our study lies in the transfer of the word2vec[22] concept from the NLP domain to bioinformatics problems, wherein genes are embedded into vectors representing diverse biological information, including their known functions in GO[31] and empirical functions in ARCHS4[32]. This has two significant implications. First, we validated that casting gene identities into their functional roles, via FRoGS, resulted in weak pathway signals within large-scale data becoming much more readily extractable (Fig. 1c).

The challenge of the weak overlap signals between two experimentally derived gene signatures is not due to the sparseness of the L1000 gene signatures, as we further tested Model $L$ based on 1000 Landmark genes alone[1] without seeing qualitative differences with respect to the extrapolated whole-transcriptome signatures. FRoGS's representation of a gene or a gene list as a numerical vector incorporating the genes' prior knowledge (Fig. 1b) is conceptually much more empowering compared to a one-hot encoding representation. Another explanation for the significantly better performance of FRoGS (Fig. 2b) is that other representations do not model GO, embed GO indirectly via text annotations, or do not utilize a GO hypergraph (Supplementary Note 1). We demonstrated by working in the biological functional space, Model $L$ outperformed the model based on NCI$_{60}$ dataset and boosted the performance of activity-based models. This highlights the benefit of both extending FRoGS vectors to incorporate knowledge derived from data sources beyond ARCHS4 and integrating Model $L$ with other experimentally generated datasets. Such sources include the plethora of gene signatures collected in The Cancer Genome Atlas, Gene Expression Omnibus, Human Protein Atlas, Functional Annotation of Mammalian Genome, Single Cell Expression Atlas, etc. Secondly, pre-trained gene functional embedding enabled transfer learning. Knowledge of genes' functions are conceptually advantageous for gene signature-based analyses, such as those that identify disease-causal genes. However, identity-based machine learning models must gain such knowledge de novo, and thus require large-scale training data that are not readily available. Using FRoGS vectors as a starting point to encode genes or gene signatures, application-specific model training requires much fewer deep-learning model parameters or can leverage non-deep-learning approaches, such as SVM[67], random forest[68], and XGBoost[69] to achieve higher performance. The ability to use small training sets is a desirable property for most bioinformatics machine-learning applications. Thus, FRoGS vectors are widely applicable to biomedical problems beyond drug target identification

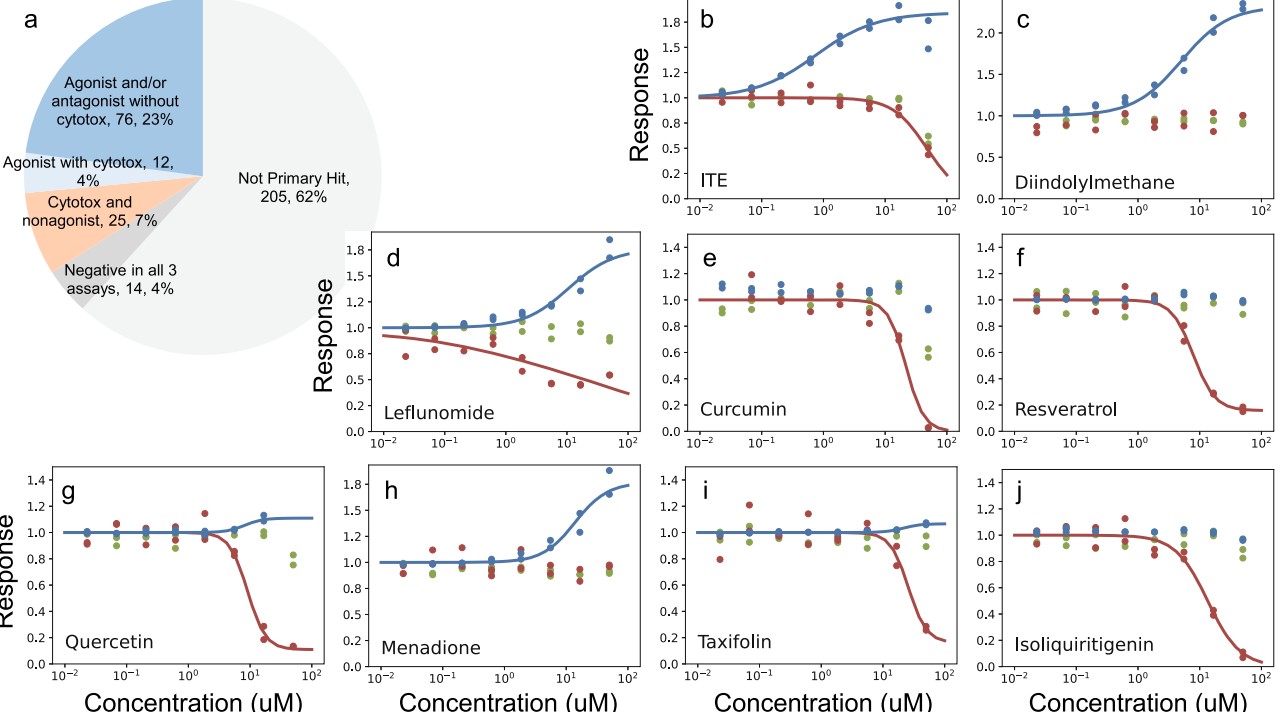

**Fig. 6 | Validation of AhR binders. a** The confirmation results of 333 compounds predicted to target AhR. **b–j** Dose-response data for selected AhR binders. Assays are color coded as blue for agonist, red for antagonist, and green for toxicity. X axes are concentrations in μM and Y axes are normalized response, with the agonist signals scaled for the ease of visual comparison.

including identifying risk genes of diseases[27], uncovering disease-disease relationships[70], and single cell clustering and classification[71].

We hypothesize one reason that preventing deep learning models from further enhancing the target recall relates to the fact that both transcriptional profiling data and compound activity profiles generally capture the downstream effect of a compound/shRNA/cDNA on its target pathway. When a protein target is modulated, downstream proteins are affected. Models based on gene expression levels cannot easily distinguish the target itself from its downstream neighbors as they show similar differential expression patterns. That is a common challenge for all transcription-based models including our Model $L$ and other activity-based models, and it contributes to the polypharmacology of the target predictions. While the target count in the positive training dataset is 1.6 genes/compound, the count in the predicted network is 4.5 genes/compound, even when only considering the high-quality subset. To overcome this, models need to further distinguish the regulation inflection point along the affected pathways by either incorporating structure-based docking simulation or narrowing the candidate pool by focusing on the densely connected protein nodes within the gene signatures. Therefore, a future direction is to scale the FRoGS framework to extract protein-protein interaction (PPI) signals in gene signatures by leveraging graph neural networks[72], similar to some recent reports[3,29].

Lastly, we demonstrated FRoGS helped predict and rescue chemical probes, as 57% of the network edges from the combined model, 45% kinase inhibitors, and 50% of AhR ligands would have been missed by using Model pQSAR alone. Going beyond the application of target deconvolution for chemical hits identified through cellular drug discovery screens, we anticipate the conceptual advance of FRoGS can replicate some of the success word2vec generated in NLP and contribute to effective bioinformatics machine learning modeling across a broad range of applications.

## Methods

### Datasets

pQSAR compound activity matrix is Novartis' internal in silico activity dataset for 5.5 million compounds over ~12,000 assays. Only assays with prediction quality exceeding the standard success threshold, $r_{ext}^2 \geq 0.3$ were retained for this study, and the predicted $Z$-scores were used to represent compound activities. Among all the perturbagens and reference compounds, 1601 have pQSAR activity profiles across 4420 assays mapped into 1003 unique gene targets. Promoter Signature Profiling (PSP) dataset consists of 18 $GR_{50}$ activity data profiles aggregated across 41 original reporter gene assays[45]. Among all the perturbagens and reference compounds, 93 have PSP profiles with at least one $GR_{50}$ value greater than 0.3, and 321 have $NCI_{60}$ profiles.

The sources of the public datasets used in this study are described in Data Availability. The L1000 Connectivity Map dataset includes both the original gene expression data for the landmark genes as well as extrapolated data for the whole transcriptome consisting of 10174 genes. 238,522 human RNA-seq samples were collected by the ARCHS4[32] database as of December 2020. Gene counts for each sample were quantified by ARCHS4 against the GRCh38 human reference genome using Kallisto[73]. Following the ARCHS4 workflow, gene counts were processed by log2 transformation and quantile normalization. The resultant gene counts were $Z$-score-normalized across samples. Genes were ranked based on their $Z$-scores within each sample and a set of differentially expressed genes were defined as genes ranked in the top 100 or bottom 100 and with $|Z| \geq 2$.

### Embeddings of individual genes encoding functional information

Two $d$-dimensional vector representations (embeddings) were created for each gene, one for GO and one for RNA-seq information. In the embedding space, functionally similar genes are embedded closely together. The same algorithm was used to learn both embeddings. Specifically, given a gene $u$, we sample a set of neighbor genes that commonly appear in the same GO processes (or RNA-seq samples) with $u$, denoted as $N(u)$, and encourage gene $u$ to have similar embeddings with its neighbor genes. To efficiently sample $N(u)$, we convert gene-GO (gene-sample) associations into a hypergraph in which, each gene is represented as a node and each GO term (RNA-seq sample) is represented as a hyperedge $e$ that connects all the genes associated with the GO term (regulated gene set of the RNA-seq sample). We then perform a random walk with restart (RWR) in the hypergraph to identify genes that lie close to each other in the graph.

For the GO graph, we assign each hyperedge $e$ a weight by the information content of the corresponding GO term[30], where more specific terms get higher weights:

$$w(e) = -\log_2 \frac{\sum_{c \in C(e)} |c|}{\sum_{e' \in E} |e'|}. \qquad (1)$$

$\frac{\sum_{c \in C(e)} |c|}{\sum_{e' \in E} |e'|}$ is the frequency of the GO term $e$, where $E$ is the set of all the GO terms included in the hypergraph, $|e|$ is the number of genes associated with the GO term $e$, and $C(e)$ is the set of all the child terms of $e$ included in the graph and $e$ itself.

For consistency, we define the weight of a hyperedge $e$ in the RNA-seq graph as:

$$w(e) = -\log_2 \frac{|e|}{\sum_{e' \in E} |e'|}, \qquad (2)$$

where $E$ is the set of all the RNA-seq samples, and $|e|$ is the number of differentially regulated genes within sample $e$.

The random walk in a hypergraph can be interpreted as: given the current node $u \in V$ (all the genes), first choose a hyperedge $e$ over all the hyperedges incident to $u$ with the probability proportional to $w(e)$, and then randomly choose a node $v \in e$ uniformly[74]. Let $\mathbf{B}$ denotes the transition probability matrix of random walk in the hypergraph, each entry of $\mathbf{B}$ is thus computed as:

$$\mathbf{B}(u,v) = \sum_{e \in E} w(e) \frac{\mathbf{H}(u,e)}{d(u)} \frac{\mathbf{H}(v,e)}{|e|}, \qquad (3)$$

where $\mathbf{H}$ is a $|V| \times |E|$ matrix with entries $\mathbf{H}(u,e) = 1$ if $u \in e$ and 0 otherwise, $d(u)$ is the degree of gene $u$ in the hypergraph, defined as $d(u) = \sum_{\{e \in E | u \in e\}} w(e)$. The RWR from a node $u$ is then defined as follows in matrix notation:

$$\mathbf{s}_u^{t+1} = (1-q)\mathbf{B}\mathbf{s}_u^t + q\mathbf{a}_u, \qquad (4)$$

where $q$ is the probability of restart, $\mathbf{a}_u$ represents the initial state, which is a $|V|$-dimensional vector with one on the $u$-th element and zeros elsewhere, $\mathbf{s}_u^t$ is a $|V|$-dimensional distribution vector that holds the probability of each node being visited after $t$ steps starting from node $u$. The distribution vectors of all the nodes form a matrix $\mathbf{S}^t$. After iterative updates, we get the stationary distribution matrix $\mathbf{S} = \mathbf{S}^\infty$ when the Frobenius norm of the difference between $\mathbf{S}^{t+1}$ and $\mathbf{S}^t$ is smaller than a predefined threshold ($10^{-6}$). A higher probability in the stationary distribution matrix indicates two corresponding genes lie closer in the hypergraph, which suggests that they share specific functions with each other compared to other genes in the graph.

For training, we adopt the contrastive learning idea, which has also been applied to learning enzyme function embeddings[75]. For each gene $u$, we sample $N(u)$ from other genes in the genome with probabilities proportional to the stationary distribution $\mathbf{s}_u$. We also sample

a set of negative samples $R(u)$ with probabilities inversely proportional to its stationary distribution $\mathbf{s}_u$. In our experiment, the ratio between the sizes of $N(u)$ and $R(u)$ is 1:5. Then the overall learning objective is to minimize the following negative loglikelihood:

$$-\sum_{u \in V} \left( \sum_{v \in N(u)} \log \sigma(\mathbf{x}_v^T \mathbf{x}_u) - \sum_{z \in R(u)} \log \sigma(\mathbf{x}_z^T \mathbf{x}_u) \right), \qquad (5)$$

where $\sigma$ is the sigmoid function. The task is thus formulated as distinguishing the functionally similar gene pairs $(u,v)$ from functionally dissimilar gene pairs $(u,z)$ through optimizing their embeddings $\mathbf{x}$.

The method is applied to two kinds of functional information to get two 256-dimension embeddings of each gene. For a gene $u$, we concatenate its two embeddings, $\mathbf{x}_u^{GO}$ and $\mathbf{x}_u^{RNA-seq}$, as a joint vector representation $\mathbf{x}_u = [\mathbf{x}_u^{GO}, \mathbf{x}_u^{RNA-seq}]$.

### Embeddings of gene signatures

Due to stochastics in biological signals and different choices of parameters in data preprocessing steps[1], "noise genes," which are genes that do not directly associate with the pathways underlying the phenotype of interest, may be included in the discovered gene set (non-pathway genes in Fig. 1a). We thus compute the consensus embedding by a linear combination of the embeddings of genes in the set, which are assigned different weights. Specifically, for a given gene $u$ in an input gene set $G$, we compute its average similarities with other genes within the set based on two kinds of embeddings, denoted as:

$$r_u^{GO} = \frac{1}{|G|} \sum_{v \in G} \cos(\mathbf{x}_u^{GO}, \mathbf{x}_v^{GO}), \qquad (6)$$

$$r_u^{RNA-seq} = \frac{1}{|G|} \sum_{v \in G} \cos(\mathbf{x}_u^{RNA-seq}, \mathbf{x}_v^{RNA-seq}), \qquad (7)$$

where $cos$ is the cosine similarity. $r_u^{GO}$ and $r_u^{RNA-seq}$ are then Z-score-normalized against the similarity distribution observed between the gene $u$ and all the other genes in the genome, denoted as $z_u^{GO}$ and $z_u^{RNA-seq}$. The weight of gene $u$ in the gene set $G$ is then limited to the range [0, 1], determined by $w_u = \min(\max(z_u^{GO}, z_u^{RNA-seq}, 0), 1)$, and the consensus embedding is computed as $\mathbf{x}_G = \frac{1}{|G|} \sum_{u \in G} w_u^* \mathbf{x}_u$, by which the impact of outlier genes in the set will be reduced. This aggregation scheme is conceptually similar to the self-attention idea that recently gained popularity in deep learning[76]; however, we used rule-based aggregation instead of a transformer neural network here because the trained transformer will only be specific to the particular training set whereas the rules crafted here can be generalized to other bioinformatics applications relying on gene set signatures.

The weight values of genes in most gene signatures of L1000 come from two types of information, while in a small portion of gene signatures, the gene weights are dominated by one type of information (Supplementary Table 4), which illustrates the complementary role of the two types of information. Intriguingly, gene signatures where RNA-seq representation dominates the weight values may suggest perturbations in pathways that are not yet discovered and captured by Gene Ontology (GO).

### Statistical analysis

**t-SNE projection.** Given a gene associated with a GO process $G$, we compute the mean $\mu$ and the standard error $\delta$ for its one-nearest neighbor (1-NN) to share the same GO process. Z-score is defined as $(\mu - \mu_0)/\delta$, where $\mu_0$ is the percentage of genes in the genome that associated with $G$. For example, among the 2476 genes associated with "immune system process", the average fraction of their 1-NNs to be also associated with the same process is $0.57 \pm 0.01$, which is higher

than the probability that the "immune system process" occurs by chance ($\mu_0 = 0.14$) with a Z-score of 43. All four GO processes examined in Fig. 1b have $p$-values $< 1^{-100}$.

**Confirmation rates.** If a confirmation rate $R_c$ was calculated based on $n$ samples, its standard error is estimates as $\sqrt{R_c(1 - R_c)/n}$. This applies to Fig. 5a–b and Supplementary Fig. 10.

### The simulation of experimental gene signatures

For a given pathway, we generated three gene sets, $S_{fg}$, $S'_{fg}$, and $S_{bg}$, each with 100 genes. $S_{fg}$ and $S'_{fg}$ were independently generated foreground gene sets, both containing $\lambda$ genes randomly sampled from pathway-associated genes and the remaining genes were uncorrelated. Thus $S_{fg}$ and $S'_{fg}$ simulated experimentally derived gene sets that resulted from two perturbagens co-targeting a common underlying pathway, where $\lambda$ controlled the level of pathway enrichment, with smaller $\lambda$ representing weaker pathway signals (pathway $W$ in Fig. 1a). $S_{bg}$ was a randomly generated background gene set. The distributions of cosine similarity between gene set embeddings were computed between the foreground-foreground pairs ($S_{fg}$, $S'_{fg}$) and foreground-background pairs ($S_{fg}$, $S_{bg}$).

### Siamese neural network architecture and balanced training

The Siamese neural network is a binary classifier that takes compound gene set embedding and shRNA/cDNA gene set embedding as a paired input and outputs the probability that the input pair are co-targeting. The neural network contains two components described as follows:

**Siamese feature extraction component.** Due to the symmetric roles of two gene sets, we designed a Siamese neural network component[77] that uses the same weights to process and extract hidden features from the compound gene signature embedding and shRNA/cDNA gene signature embedding by a single layer fully connected neural network:

$$D(\mathbf{x}) = Dense_1(\mathbf{x}), \qquad (8)$$

where $\mathbf{x}$ is the input FRoGS gene signature embedding. Following a standard operation for modeling the symmetric relations of two inputs[78–80], the outputs are element-wise multiplied as $D(\mathbf{x}_{cpd}) \odot D(\mathbf{x}_{gene})$ and fed into the classification component, which is a binary operation that takes in two vectors of the same dimensions and returns a vector of the multiplied corresponding elements.

**Classification component.** We built a two-layer fully connected neural network as the classifier. The sigmoid activation function is applied in the last layer, which produces a scalar output in the range [0, 1]:

$$o = Dense_3(Dense_2(D(\mathbf{x}_{cpd}) \odot D(\mathbf{x}_{gene}))), \qquad (9)$$

We use the cross-entropy loss for the training of the neural network:

$$-\sum_i (y_i \log o_i + (1 - y_i) \log(1 - o_i)), \qquad (10)$$

where the binary label $y_i$ indicates whether the input gene is the true target of the input compound.

The quantities of neurons in the hidden layers are hyperparameters. In our experiment, we used 2048 and 512 neurons for the $Dense_1$ and $Dense_2$ layers, respectively. A balanced training strategy was applied to train a model that is not biased towards popularly known targets. Specifically, if a gene appears in a positive pair, we sample a compound that is not annotated to target this gene to form a negative pair with it. That is, for each gene, the ratio of positive to

negative pairs is 1:1. This is critical for ensuring the generalization of the model as detailed in Supplementary Note 2.

The training and test set were split by compounds, where compounds with known targets were placed into the training set while the other compounds profiled in L1000 were placed into the test set. Neural network models were trained for pairs of compound and shRNA signatures, as well as pairs of compound and cDNA signatures. In the model involving compound and shRNA signatures, the positive dataset comprised 2308 compound-target pairs for 1424 compounds in 12 cell lines, resulting in 12,332 positive gene signature pairs when considering the different combinations of compounds, targets, and cell lines. In the case of the model with compound and cDNA signatures, the positive dataset consisted of 1473 compound-target pairs for 1061 compounds in 10 cell lines, resulting in 9800 positive gene signature pairs across different combinations. The number of negative gene signature pairs was sampled to match the number of positive pairs for each model. With the trained models, we performed inference for compound-target gene pairs using cDNA gene signatures and shRNA gene signatures respectively for 20,306 compounds and 4784 genes across 13 cell lines, resulting in predictions for a total of 393,525,524 combinations, which contains 73,380,830 unique compound-target gene pairs. Performance of the different models was evaluated using fivefold cross-validation on the training set. For each evaluation, we trained three models on the same data, and the average value of the predicted probabilities from the three models was computed for each $(c, g)$ pair and used for target ranking. Each model underwent training for 60 epochs. For models using shRNA signatures, the average training time for each model was 93 s across five models. For models using cDNA signatures, the average training time was 82 s across five models. A K80 GPU was used for model training and inference.

### Similarity calculation

The structure similarity score of a compound pair, $s$, was calculated based on the Tanimoto score using their ECFP4 chemical fingerprints[81]. The mapping between $s$ and SAR probability value, $p(s)$, was computed based on how often compound pairs with the given similarity were found co-active in the same biological assay according to Novartis' in-house database (Supplementary Note 4). This mapping is also labeled as Model $S_{SAR}$. For compounds with activity profiles including pQSAR, $NCI_{60}$, and PSP, the Pearson correlation coefficient across their profile vectors, $q$, was used as the similarity score.

### Consensus input gene set signatures

The expansive collection of gene signatures associated with the L1000 dataset is based on treatments using genomic reagents (both loss-of-function shRNA and gain-of-function cDNA treatments for each gene $g$) and compounds $c$ dosed at multiple concentrations and durations screened across multiple cell lines. Therefore, given a specific compound-target pair, $(c, g)$, multiple choices of input signatures exist for $c$ and $g$, individually as well as combinatorically. We reduced this ambiguity by first consolidating all versions of compound signatures, retaining only one exemplar gene set signature per cell line that had the highest Transcriptional Activity Score $(TAS)^{[1]}$. For model input, only the shRNA/cDNA signature $g$ acquired under the same cell line was paired with each compound to form a $(c, g)$ vector pair.

### Consensus predicted target ranking

For L1000-based target prediction, an established convention in the field for algorithm performance comparison is to calculate the percentage of compounds having their known target recalled, when a certain top percentage of target candidates is considered. We thus chose to combine multiple target prediction lists in a way that best reflected this practice. A target list for a given compound contained $n$ candidate genes that were first sorted based on their decreasing

probability scores, and then their rank order $O_g$ was identified and normalized as $O_g/n$. All target candidates for the same compound were then pooled, genes were sorted based on their normalized ranks, and the first occurrence of a candidate $g$ was retained yielding a unique consensus target list for compound $c$. This rank-based multi-list aggregation method was found to be more effective in target recall with value 0.36 at the top 5% compared to 0.12 obtained by pooling and sorting gene candidates based on their model-predicted probability scores.

### Logistic regression modeling

Given a feature vector $\mathbf{x}_i$, logistic regression (LR) model estimates the probability of the record being positive as:

$$p(\mathbf{x}_i) = \frac{1}{1 + \exp\{-b - \mathbf{w}\mathbf{x}_i\}}, \tag{11}$$

where $\mathbf{w}$ is a weight vector and $b$ is a bias. Feature definition depends on the data source. For compound-target associations predicted by our deep learning model, $r(c, g)$ is the normalized consensus rank of gene $g$ in the compound's putative target list, and feature $x$ is defined as the odd ratio:

$$x = \frac{1 - r}{r}. \tag{12}$$

For the compound activity profile matrix, $q$ was the Pearson correlation coefficient of the pharmacological profiles between a query compound $c$ and a reference compound $c'$. Considering multiple reference compounds $\{c'_i\}$ may exist for a target $g$, the maximum value of $q$ was used as the feature:

$$x = \max_i q(c, c'_i). \tag{13}$$

When $NCI_{60}$ or PSP activity profiles were used for modeling, $q$ refers to the Pearson correlations in the corresponding activity matrices. In the combined Model, $\mathbf{x}_i$ consists of two logits computed from Model $L$ and Model pQSAR, respectively. Distinct from the standard LR model, we require weight $\mathbf{w}$ to be non-negative to ensure the monotonic relationship between $p$ and $\mathbf{x}$, ensuring compliance with the biological guilt-by-association (GBA) principle.

The LR cost is defined as the combination of cross entropy and an $L_2$ regularization term with sampling weighting:

$$\mathcal{L} = \frac{1}{n_p} \sum_{i=1}^{n_p} \log(p(\mathbf{x}_i)) + \frac{1}{n_n} \sum_{i=1}^{n_n} \log(1 - p(\mathbf{x}_i)) + \lambda \|\mathbf{w}\|^2, \tag{14}$$

where $\lambda$ is the only regularization hyperparameter.

Model training used a standard nested cross-validation loop, where the inner loop searched for the optimal $\lambda$ and the outer loop evaluated model performance with five-fold cross validations.

The positive training dataset consisted of 2340 $(c, g)$ pairs formed between 1438 compounds and 499 targets. A total of 5,614,766 $(c, g)$ combinations of the 1438 compounds with genes that are not annotated as targets formed the negative training dataset. During the training, we randomly sampled 231,660 negative pairs, so that the final training set consists of 1% positive labels and 99% negative labels. The positive and negative labels made equal contributions to the model training after weighting. The test dataset consisted of 67,763,724 $(c, g)$ pairs formed between 18,855 compounds and 4784 targets, which included 2433 $(c, g)$ pairs formed by 509 polypharmacological compounds ( > 5 targets/compound) and 507 targets. As Model $L$, Model pQSAR, and Model $LQ$ contained only 2, 2, and 3 parameters, respectively, model training were robust against random seeds and only cost a few seconds.

**Imbalanced dataset correction for logistic regression models**

Broad's compound-gene dataset was highly imbalanced during the logistic regression model training, as only a small fraction was annotated as a known association and assigned positive labels, while the majority had no curation and were assigned negative labels. Performance metrics such as accuracy and area under the curve (AUC) for receiver operating characteristic (ROC) can be excessively optimistic for such imbalanced domains[82]. Precision-recall curves reduce but do not eliminate the effect of sample imbalance. When models relying on different features were trained, the size and composition of training datasets varied due to different number of records containing missing features required by different models. Performance metrics for different models were therefore associated with different sample sizes and needed to be normalized for comparison. In model training, we always assigned sample weights, so that positive and negative labels were normalized into an effectively balanced (1:1) dataset.

Specifically, if there were $n_p$ positive samples and $n_n$ negative samples, sample weights were $1/n_p$ and $1/n_n$, respectively. Sample counts for true positive (TP), false positive (FP), true negative (TN), and false negative (FN) were also corrected with the sample weights by multiplying $1/n_p$, $1/n_n$, $1/n_n$ and $1/n_p$, respectively. Performance metrics-such as precision, recall, F1, Matthews Correlation Coefficient (MCC), AUC of ROC or precision-recall (PR) curve, etc., were adjusted accordingly by the same sample weighting scheme so that they characterize the model as if all training datasets were in balance. Metrics for all models studied are provided in Supplementary Table 1.

**Biochemical HTRF kinase activity inhibition assays**

All biochemical kinase activity assays were carried out using the TR-FRET method as described by Yong et al.[83]. Kinase reactions were conducted in white or black 1536-well untreated microplates (Greiner Bio-One; Monroe, NC) with conditions for each reaction listed in Supplementary Data 4. ATP concentrations used were around the apparent ATP $K_m$ value for each enzyme. The enzymes were first pre-incubated in the presence of compounds for 15 min prior to the addition of the appropriate substrate and ATP to reach a final reaction volume of 4 μL and a final DMSO concentration of 0.5%. At a designated time point for each assay, the reactions were quenched by additional of equal volumes of a 200 mM EDTA solution and a HTRF detection solution, consisting of a Europium-cryptate-conjugated antibody and SA-XL665 (610SAXLB) (both from PerkinElmer; Beford, MA) diluted in a detection buffer (400 mM KF, 50 mM HEPES, pH 7.0, 0.1% BSA, 0.01% Tween20), bringing the final volume to 8 μL. For reactions using KinEASE kit and LANCE Ultra kit, the detections were done following manufacturer's protocols. After addition of the detection reagents, the plates were incubated at RT for 1 h, then read using a PHERAstar FSX equipped with a compatible TR-FRET module (BMG Labtech; Cary, NC).

Compound activity was initially assayed with a single final concentration of 50 μM in triplicate, and the percentage of inhibition of each compound was calculated. Hits were selected for confirmation in 8-point dose-response assays in duplicate, with a top concentration of 50 μM and a 1:3 serial dilution. $IC_{50}$ values were defined as the inhibitor concentration at which 50% of the enzyme activity is inhibited. All compound dispensing was performed using an Echo 555 acoustic liquid handler (Beckman Coulter Life Sciences; Indianapolis, IN).

**AhR reporter gene assays and CellTiter-Glo cytotoxicity assay**

HepG2-Lucia AhR (a human HepG2 hepatoma-derived cell line stably expressing the secreted Lucia luciferase reporter gene under the control of a minimal promoter coupled with the human Cyp1a1 gene's entire regulatory sequence) was purchased from InvivoGen

(San Diego, CA) with catalog code hpgl-ahr. The cell line was maintained in Gibco Minimum Essential Medium (MEM) supplemented with 10% fetal bovine serum (FBS) (Avantor; Radnor Township, PA), 1% Cytiva HyClone MEM non-essential amino acids (NEAA) 100X Solution, 1% Gibco antibiotic-antimycotic (Anti-Anti) 100X and 100μg/mL of Zeocin. All incubations were conducted at 37 °C with 5% $CO_2$ unless otherwise stated. Medium and other supplements were purchased from Thermo Fisher Scientific (Waltham, MA).

All assays were conducted in custom 1536-well, white, solid-bottom, tissue culture-treated assay plates (#789173-A, Greiner Bio-One; Monroe, NC). For the single-point primary screens, the wells were pre-spotted with 25nL of compounds in DMSO (50 μM final concentration) using an Echo 555 acoustic liquid handler (Beckman Coulter Life Sciences; Indianapolis, IN). A total of 2500 cells in a final 5 μL were seeded in each well in above described medium and supplements, except the Zeocin. The AhR reporter gene assays were carried out as both agonist and antagonist assays. For the agonist assay, plates were incubated for 24 h. For the antagonist and the cytotoxicity assay, plates were first incubated for 30 min, after which a final concentration of 15 nM of 2,3,7,8-Tetrachlorodibenzo-p-dioxin (TCDD) (AccuStandard; New Haven, CT) was added to induce reporter gene expression followed by an incubation of 24 h and 48 h, respectively. After incubation, 2.5 μL per well of a coelenterazine-based luminescence assay reagent QUANTI-Luc (InvivoGen; San Diego, CA) or CellTiter-Glo (Promega; Madison, WI) was dispensed into the plates for both reporter gene assays and the cytotoxicity assay, respectively. Luminescence signals were read on the Luminescence Plate Reader (LPR) (#1222-9001X1, GNF Systems; San Diego, CA) immediately following the QUANTI-Luc addition or after a 10 min incubation at room temperature following the CellTiter-Glo addition. Hit compounds from the single-point primary screens were selected for dose-response characterizations using the same assays and conditions described above. Compound hits were re-arrayed and pre-spotted in the format of 8-point, 1:3 serial dilution with a top final concentration of 50 μM.

**Reporting summary**

Further information on research design is available in the Nature Portfolio Reporting Summary linked to this article.

## Data availability

pQSAR activity dataset and PSP dataset are large-scale proprietary Novartis in-house resources, which cannot be released due to confidentiality restrictions. All other datasets were from the public domain. The level-5 perturbagen profiles for the L1000 Connectivity Map dataset were downloaded from the Gene Expression Omnibus (https://www.ncbi.nlm.nih.gov/geo) under Accession IDs GSE92742 and GSE70138. The Broad's compound-target annotations were extracted from the Connectivity Map website (https://clue.io) using the developer API. The Reactome pathway data were downloaded from https://reactome.org. Gene Ontology data were downloaded from http://geneontology.org and processed by Metascape (https://metascape.org). ARCHS4 gene expression profiles were obtained from https://maayanlab.cloud/archs4 with the data file link https://s3.amazonaws.com/mssm-seq-matrix/human_matrix.h5. The US National Cancer Institute 60 human tumor cell line anticancer drug screen ($NCI_{60}$) dataset were downloaded from https://discover.nci.nih.gov/cellminer. The experimental data for kinase binders and AhR binders can be found in Supplementary Table 2-3 and Supplementary Data 2-3. After removing 1163 compound-target pairs that are validated only according to in-house activity database, 5133 out of the 6296 predictions with multiple lines of additional validation evidence are made available as a community resource in Supplementary Data 1. Source data are provided with this paper.

## Code availability

Custom Python code and the trained gene embedding vectors are made available at https://github.com/chenhcs/FRoGS.

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

## Acknowledgements

This work was partially supported by the NIAID grants 5U19AI135972 and 75N93019C00046, DoD grant W81XWH-20-1-0270 to S.K.C.

## Author contributions

H.C. developed prediction models. H.C., Y.Zhong, B.Z., F.J.K. and Y.Zhou performed data analyses. Y.W., J.H., C.J.C., J.L.W., Y.J. performed experimental validations. Y.Zhou, L.P., M.W.C., C.B. and S.K.C. con-ducted an initial proof-of-concept study. F.J.K., S.K.C., T.J., J.J. and Y.Zhou directed the project. H.C., F.J.K. and Y.Zhou wrote the manuscript.

## Competing interests

The authors declare no competing interests.

## Additional information

**Supplementary information** The online version contains
supplementary material available at

Hao Chen or Yingyao Zhou.

**Peer review information** *Nature Communications* thanks Slim Fourati,
Anil Jegga, and Louxin Zhang for their contribution to the peer review of
this work. A peer review file is available.

