## [Peer Review File · Nature Communications]

Drug target prediction through deep learning functional representation of gene signaturesReviewer #1 (Remarks to the Author):

In this manuscript, the authors report Functional Representation of Gene Signatures (FRoGS), a neural network-based framework for generating representations (embeddings) of gene signatures. The authors use functional (GO terms) and RNA-seq-based similarities as individual gene embeddings in this framework. Random walks were then used to identify gene neighborhoods and train the gene embedding vectors (similar to a word2vec Skipgram model), and two sets of embedding vectors were concatenated to generate the final gene representations. These embedding vectors were then used to generate the embeddings of gene lists (pathways or gene signatures) using weighted linear combinations, similar to a self-attention mechanism. The authors used gene signatures associated with compound and genomic perturbations (both loss-of-function shRNA and gain-of-function cDNA) from the L1000 Connectivity Map and generated signature embeddings for each compound and target. They trained a simple two-layered Siamese neural network for compound–target predictions. Compound–target associations from L1000 were used to train this model. Multiple predictions associated with the same compound–target pair (different cell lines and perturbation types) were then aggregated into (normalized) consensus ranks and converted to probability scores using a modified version of a logistic regression model. Their model displayed significantly higher known target recall rates than the other baseline models (trained using the same Siamese network). The model also outperformed other activity-based target prediction models. The authors further demonstrated the utility of combining their FRoGS model with other activity-based models to create multimodal target predictors. Finally, they performed experimental validation of the predicted kinase inhibitors and compounds targeting aryl hydrocarbon receptors (AhR).

Although the approach itself is reasonable and methodologically somewhat novel, conceptually gene- and pathway-based (or feature-based) connectivity mapping for drug discovery or drug repositioning has been performed previously. Similarly, Word2Vec has been used by several groups for various bioinformatics problems. However, the authors have done a good job with the overall implementation of their model, especially using hypergraphs and generating signature representations from the gene embeddings. However, I have a few concerns.

1. From the manuscript, it is not clear if the authors used the gene expression data for the landmark genes or the extrapolated data (BING – or best inferred genes) consisting of ~10k genes.
2. It is also unclear what the final training (both positive and negative associations) and test set sizes are. It would be relevant to have a table or at least mention it in the manuscript or supplementary text.
3. Were both models (the Siamese network and the logistic regression models) trained on the same dataset or were any additional changes made for each case? Also, how long did the training last in terms of epochs over the training sets?
4. It would be relevant to mention the dimensionality of the gene vectors (x) used in the representation learning step.
5. Discussion about the impact of gene set size (fixed at 100 genes in the current set of experiments) in the simulation experiments is warranted to verify if the performance improvements observed using FRoGS hold in the case of smaller (or larger) gene sets.
6. While generating the gene signature representations, the authors computed gene similarities (and subsequently z-scores) based on both the GO and RNA-seq representations. Because they ended up using the highest z-score value among the two, it would be interesting to see if the authors observed any single gene similarity source dominating the final weight values used in all the linear combinations associated with each signature.

7. Did the authors explore using heterogeneous graphs by combining all the different similarity sources (GO, RNA-Seq, and perhaps even more) and using graph convolutional models to identify the gene representations? This approach could not only help bring in second-order (indirect) gene neighbors into the analysis but also include more omics data.

8. Aren't the t-SNE projection figure (Fig 1) and related annotations misleading because they already use gene function-based embeddings? Therefore, isn't the claim about FROGS embedding being a proxy for functional relatedness expected and not surprising? In addition, the authors used very broad categories to generate the map.

9. The authors should consider including the methodological and data-related limitations of their approach and potential extensions of the current framework. For instance, the possibility of using single cell signatures (e.g., Human single cell atlas data) instead of or in addition to the ARCHS4 data. Likewise, limitations associated with the L1000 data. Importantly, some discussion about the real world and practical utility of this model for drug targeting should be included. For example, FROGS or similar approaches narrow down the search space; however, these models often predict several compounds that target a single protein. In the current study too, the authors have reported 369 compounds potentially targeting AhR.

10. The manuscript requires significant language editing. There are several spelling and grammatical errors throughout. The authors should also refrain from using terms such as "more effective" or "high-quality" when referring to the compound-target predictions from FROGS.

Reviewer #2 (Remarks to the Author):

What are the noteworthy results?

Chen H. et al. applied the word2vec concept to embed biological functions (from GO and RNA-Seq experiments) in genes, which can be used to assess the similarity between two perturbations. The authors show that their embedding strategy, named FROGS, outperforms other embedding methods such as OPA2Vec (pre-trained on PubMed abstract) to identify similarity in gene expression between perturbations. The authors used FROGS to predict compound-target relationships, which they confirmed experimentally in two cases. The paper is well-written, and the results for compound-target prediction are of high interest. Whether FROGS could be used for other tasks like PPI prediction or gene-disease associations has not been shown in the paper, but it looks like other potential application of FROGS.

Will the work be of significance to the field and related fields? How does it compare to the established literature? If the work is not original, please provide relevant references.

The results are in line with the literature. When there were discrepancies, ex with CMap, they were highlighted in the supplementary notes. A section on the reasons why the authors think FROGS outperforms other embedding methods to assess the similarity of two perturbations (in Figure 1) should be added to the discussion of the paper to bonify the work.

Does the work support the conclusions and claims, or is additional evidence needed?

The figures and supplementary tables do support the claims of the paper.

Are there any flaws in the data analysis, interpretation and conclusions? Do these prohibit publication or require revision?

I only minor corrections:

page 2 paragraph 2: replace "a well-established statistical framework" with "well-established statistical frameworks"

page 3 paragraph 3: replace "the same color" by "same biological functon(i.e. node color in Figure 1b)".

page 4 paragraph 2: Did the authors consider comparing FROGS with rank-rank based similarity metrics such as RRHO (Plaisier SB et al, 2010 NAR) and if embedding methods would also outshine those type methods (they should perform better than Fisher's exact test)?

page 4 paragraph 3: How many compounds had more than 5 targets and were excluded from the L1000 database?

Is the methodology sound? Does the work meet the expected standards in your field?

The methodology is sound, and the method section (and supplementary notes) provide adequate details to understand the methodology.

Is there enough detail provided in the methods for the work to be reproduced?

The Python code to rerun some of the FROGS analysis is provided in GitHub but would benefit from better walkthrough examples.

Reviewer #3 (Remarks to the Author):

In current bioinformatics research, many machine learning applications rely on one-hot-encoding strategy (i.e. gene identities only), often missing the opportunity to leverage existing knowledge about gene functions. In response to this, the authors propose a novel approach that maps genes to a high dimensional numerical vector through the functional representation of genes, named FROGS. The method reimagines gene signatures, projecting them onto their biological functions rather than their identities, akin to the way the word2vec technique operates in natural language processing.

The authors validate the proposed gene embedding using different tests against a few existing gene embeddings, including Gene2Vec, OPA2Vec-Gene and OPA2Vec-GO. First, FROGS facilitates extraction of weak pathway signals from gene signature more significantly than the other gene embeddings.

Second, the authors experimental investigation showcases the remarkable potential of FROGS when applied to L1000 datasets. It outperformed other models, resulting in significantly more accurate compound-target predictions. By seamlessly integrating additional pharmacological activity data sources, FROGS makes a substantial contribution to generating high-quality compound-target predictions.

Third, the proposed gene embedding results in good prediction methods with a small number of parameters that hence require less training data than gene-identity-based machine learning methods.

Furthermore, the authors illustrated that FROGS not only facilitated the prediction but also the recovery of chemical probes. Specifically, they found that when employing the combined model, 57% of network edges, 45% of kinase inhibitors, and 50% of AhR ligands would have been overlooked if relying solely on Model pQSAR.

Their validation studies highlight the broad applicability of FROGS in the realm of machine learning-based bioinformatics. By equipping prediction networks with prior knowledge of gene functions, we can expedite the discovery of relationships among gene signatures obtained from large-scale OMICS studies encompassing compounds, cell types, disease models, and patient cohorts.

The content presented thus far is commendable. However, there is room for enhancement in the manuscript. Throughout the document, certain sentences appear awkward and ambiguous, while a few terms require clear definitions. Detailed comments addressing these issues can be found below.

In summary, this is a valuable research. It advances gene embedding techniques. However, it needs further revision before the acceptance decision is made.

Minor comments

1. Page 2.

In the first paragraph, provide the full name of shRNA which appears for the first time.

In the second paragraph, what does "human host" mean?

2. In the first sentence of the results section "FRoGS vectors were pre-trained such that individual human genes ...", The word "pre-trained" should be "pre-training". Otherwise, how the model is pre-trained is not given. The author should describe how they pre-train the model in the Method section.

3. Page 5.

What are "orthogonal data sources"? The short explanation is necessary here, although such a term is discussed in the supplemental notes.

Fig. 1b demonstrates the proposed method leads good correlation between genes with similar functions. The author states: 'Genes closely positioned in the same clusters tend to share the same color ($p < 10^{-100}$)'. However, the visualization does not support this highly significance. Describe how to compute the p-value.

Revise "The application of Model L to both training and test datasets resulted in the prediction of 780,438 compound-target pairs with a probability above 0.8." It is confusing.

The statement "covering 1601, 93, and 321 Broad compounds, respectively" is not clear, with only 1438 drugs are used as the positive pairs in training set. 'a Validation category for 2491 Model L predictions' in Figure 3.a seems inconsistent with "2477 predictions fall into the three categories assigned based on their best supporting evidence.'

4. Page 11. What are "noise" genes?

5. Page 12

Pairwise product $D(x_{cpd}) \odot D(x_{gene})$ seems not meaningful. Given a brief explanation.

6. Page 13. "perturbagens" should be a typo.

We thank the three reviewers for their insightful technical evaluations and thoughtful comments. We hope that this revision improves the clarity, incorporates the valuable suggestions from the reviewers, and addresses the concerns raised.

The point-by-point response to reviewer's comments is provided below.

REVIEWER COMMENTS

Reviewer #1 (Remarks to the Author):

In this manuscript, the authors report Functional Representation of Gene Signatures (FRoGS), a neural network-based framework for generating representations (embeddings) of gene signatures. The authors use functional (GO terms) and RNA-seq-based similarities as individual gene embeddings in this framework. Random walks were then used to identify gene neighborhoods and train the gene embedding vectors (similar to a word2vec Skipgram model), and two sets of embedding vectors were concatenated to generate the final gene representations. These embedding vectors were then used to generate the embeddings of gene lists (pathways or gene signatures) using weighted linear combinations, similar to a self-attention mechanism. The authors used gene signatures associated with compound and genomic perturbations (both loss-of-function shRNA and gain-of-function cDNA) from the L1000 Connectivity Map and generated signature embeddings for each compound and target. They trained a simple two-layered Siamese neural network for compound–target predictions. Compound–target associations from L1000 were used to train this model. Multiple predictions associated with the same compound–target pair (different cell lines and perturbation types) were then aggregated into (normalized) consensus ranks and converted to probability scores using a modified version of a logistic regression model. Their model displayed significantly higher known target recall rates than the other baseline models (trained using the same Siamese network). The model also outperformed other activity-based target prediction models. The authors further demonstrated the utility of combining their FRoGS model with other activity-based models to create multimodal target predictors. Finally, they performed experimental validation of the predicted kinase inhibitors and compounds targeting aryl hydrocarbon receptors (AhR).

Although the approach itself is reasonable and methodologically somewhat novel, conceptually gene- and pathway-based (or feature-based) connectivity mapping for drug discovery or drug repositioning has been performed previously. Similarly, Word2Vec has been used by several groups for various bioinformatics problems. However, the authors have done a good job with the overall implementation of their model, especially using hypergraphs and generating signature representations from the gene embeddings. However, I have a few concerns.

1. From the manuscript, it is not clear if the authors used the gene expression data for the landmark genes or the extrapolated data (BING – or best inferred genes) consisting of ~10k genes.

The results presented were based on BING, as explained on page 4:

“With each compound and genomic perturbation represented by an aggregated FRoGS signature vector corresponding to their extrapolated whole-transcriptome L1000 profiles (see Methods)”

We provided the detailed gene counts for BING in Methods (page 10):

“The dataset includes both the original gene expression data for the landmark genes as well as extrapolated data for the whole transcriptome consisting of 10174 genes.”

2. It is also unclear what the final training (both positive and negative associations) and test set sizes are. It would be relevant to have a table or at least mention it in the manuscript or supplementary text.

We appreciate the suggestion. The sizes are now provided on page 13 regarding the Siamese network:

“Neural network models were trained for pairs of compound and shRNA signatures, as well as pairs of compound and cDNA signatures. In the model involving compound and shRNA signatures, the positive dataset comprised 2,308 compound-target pairs for 1,424 compounds in 12 cell lines, resulting in 12,332 positive gene signature pairs when considering the different combinations of compounds, targets, and cell lines. In the case of the model with compound and cDNA signatures, the positive dataset consisted of 1,473 compound-target pairs for 1,061 compounds in 10 cell lines, resulting in 9,800 positive gene signature pairs across different combinations. The number of negative gene signature pairs was sampled to match the number of positive pairs for each model. With the trained models, we performed inference for compound-target gene pairs using cDNA gene signatures and shRNA gene signatures respectively for 20,306 compounds and 4,784 genes across 13 cell lines, resulting in predictions for a total of 393,525,524 combinations, which contains 73,380,830 unique compound-target gene pairs.”

On page 15, we detailed the numbers for the logistic regression models:

“The positive training dataset consisted of 2340 (c, g) pairs formed between 1438 compounds and 499 targets. A total of 5,614,766 (c, g) combinations of the 1438 compounds with genes that are not annotated as targets formed the negative training dataset. During the training, we randomly sampled 231,660 negative pairs, so that the final training set consists of 1% positive labels and 99% negative labels. The positive and negative labels made equal contributions to the model training after weighting. The test dataset consisted of 67,763,724 (c, g) pairs formed between 18,855 compounds and 4784 targets, which included 2433 (c, g) pairs formed by 509 polypharmacological compounds (>5 targets/compound) and 507 targets.”

3. *Were both models (the Siamese network and the logistic regression models) trained on the same dataset or were any additional changes made for each case? Also, how long did the training last in terms of epochs over the training sets?*

The difference between the training datasets were detailed in our reply to the previous comment.

On page 13, we added training details for the Siamese network models:

“Each model underwent training for 60 epochs. For models using shRNA signatures, the average training time for each model was 93 seconds across five models. For models using cDNA signatures, the average training time was 82 seconds across five models. A K80 GPU was used for model training and inference.”

On page 15, we added training details for the logistic regression models:

“As Model *L*, Model pQSAR, and Model *LQ* contained only 2, 2, and 3 parameters, respectively, model training were robust against random seeds and only cost a few seconds.”

4. *It would be relevant to mention the dimensionality of the gene vectors (x) used in the representation learning step.*

Added to page 11:

“The method is applied to two kinds of functional information to get two 256-dimension embeddings of each gene.”

5. Discussion about the impact of gene set size (fixed at 100 genes in the current set of experiments) in the simulation experiments is warranted to verify if the performance improvements observed using FRoGS hold in the case of smaller (or larger) gene sets.

We carried out simulations using 50 genes and 200 genes and the conclusions remained unchanged.

We updated the Results (page 4):

“More detailed analyses using pathway R-HSA-5576891 (cardiac conduction) as an example are provided in Fig. S1a, further supporting the conclusion presented in Fig. 1c, even as the size of gene lists vary (Fig. S1b-c).”

The details are provided as two new figures in Fig. S1b and Fig. S1c:

Fig. S1b

Fig. S1c

b-c Simulation results validate the results presented in Fig 1c and are robust against gene lists of various sizes, where the same percentages of pathway genes used in Fig 1c (5%, 10%, 15%, and 20%) seed the foreground gene sets. **b** Results obtained with $\lambda = 3$, $\lambda = 5$, $\lambda = 8$, and $\lambda = 10$ for

gene sets of size 50. **c** Results obtained with $\lambda = 10$, $\lambda = 20$, $\lambda = 30$, and $\lambda = 40$ for gene sets of size 200.

6. While generating the gene signature representations, the authors computed gene similarities (and subsequently z-scores) based on both the GO and RNA-seq representations. Because they ended up using the highest z-score value among the two, it would be interesting to see if the authors observed any single gene similarity source dominating the final weight values used in all the linear combinations associated with each signature.

We thank the reviewer for raising this interesting question.

We added on page 12:

“The weight values of genes in most gene signatures of L1000 come from two types of information, while in a small portion of gene signatures, the gene weights are dominated by one type of information (Table S7), which illustrates the complementary role of the two types of information. Intriguingly, gene signatures where RNA-seq representation dominates the weight values may suggest perturbations in pathways that are not yet discovered and captured by Gene Ontology (GO).”

Data are available as Table S7:

	Dominated by GO	Dominated by RNA-seq
In Compound signatures	1.29%	8.75%
In shRNA signatures	7.59%	18.87%
In cDNA signatures	2.17%	10.85%

7. Did the authors explore using heterogeneous graphs by combining all the different similarity sources (GO, RNA-Seq, and perhaps even more) and using graph convolutional models to identify the gene representations? This approach could not only help bring in second-order (indirect) gene neighbors into the analysis but also include more omics data.

On page 9, we added the discussion on using graph neural network as a future direction:

“To overcome this, models need to further distinguish the regulation inflection point along the affected pathways by either incorporating structure-based docking simulation or narrowing the candidate pool by focusing on the densely connected protein nodes within the gene signatures. Therefore, a future direction is to scale the FROGS framework to extract protein-protein interaction (PPI) signals in gene signatures by leveraging graph neural networks⁷², similar to some recent reports^{3,29}.”

We would like to point out that indirect gene neighbors have already been taken into account during the current FROGS vector training in the form of the stationary matrix, where the closeness of two genes were extracted through an iterative random walk process conceptually similar to the message passing process in graph neural networks (described on page 11).

8. Aren't the t-SNE projection figure (Fig 1) and related annotations misleading because they already use gene function-based embeddings? Therefore, isn't the claim about FROGS embedding being a proxy for functional relatedness expected and not surprising? In addition, the authors used very broad categories to generate the map.

The reviewer is correct that the t-SNE projection is not a discovery; the purpose of the t-SNE plot is to serve as a visual confirmation that our trained embedding vectors indeed captured gene functions. To further clarify this point, we made the following modification (underlined) (page 3):

“For the purpose of visually confirming the validity of the FROGS vectors, we used a 2-dimensional t-SNE projection (Fig. 1b) to confirm whether individual genes were grouped based on their functions in the embedding space, in a manner similar to how synonyms are co-located in the word2vec embedding.”

The locations of the genes on the t-SNE map were generated using the FROGS vectors alone, not relying on any ontology categories. The four ontology categories were only used to color the markers. Human genes are annotated by over 15,000 hierarchical ontology categories. We chose four high-level ontology terms, i.e., only four distinct colors, to make it easy for readers to appreciate the functional colocation pattern. Using more granular ontology terms will introduce too many colors and make the visualization difficult to interpret.

9. The authors should consider including the methodological and data-related limitations of their approach and potential extensions of the current framework. For instance, the possibility of using single cell signatures (e.g., Human single cell atlas data) instead of or in addition to the ARCHS4 data. Likewise, limitations associated with the L1000 data. Importantly, some discussion about the real world and practical utility of this model for drug targeting should be included. For example, FROGS or similar approaches narrow down the search space; however, these models often predict several compounds that target a single protein. In the current study too, the authors have reported 369 compounds potentially targeting AhR.

We thank the reviewer for the excellent suggestions. We made a few revisions in the Discussion.

On page 9 (changes underlined):

“This highlights the benefit of both extending FROGS vectors to incorporate knowledge derived from data sources beyond ARCHS4 and integrating Model L with other experimentally generated datasets. Such sources include the plethora of gene signatures collected in The Cancer Genome Atlas, Gene Expression Omnibus, Human Protein Atlas, Functional Annotation of Mammalian Genome, Single Cell Expression Atlas, etc.”

We have now added the following paragraph to Discussion (page 9) regarding the limitations of the L1000 data. The challenge of models’ tendency in predicting many targets per compound is discussed in more detail.

“We hypothesize one reason that preventing deep learning models from further enhancing the target recall relates to the fact that both transcriptional profiling data and compound activity profiles generally capture the downstream effect of a compound/shRNA/cDNA on its target pathway. When a protein target is modulated, downstream proteins are affected. Models based on gene expression levels cannot easily distinguish the target itself from its downstream neighbors as they show similar differential expression patterns. That is a common challenge for all transcription-based models including our Model L and other activity-based models, and it contributes to the polypharmacology of the target predictions. While the target count in the positive training dataset is 1.6 genes/compound, the count in the predicted network is 4.5 genes/compound, even when only considering the high-quality subset. To overcome this, models need to further distinguish the regulation inflection point along the affected pathways by either

incorporating structure-based docking simulation or narrowing the candidate pool by focusing on the densely connected protein nodes within the gene signatures. Therefore, a future direction is to scale the FROGS framework to extract protein-protein interaction (PPI) signals in gene signatures by leveraging graph neural networks⁷², similar to some recent reports^{3,29}.”

We agreed with the reviewer that “some discussion about the real world and practical utility of this model for drug targeting should be included.” We also would like to point out that the discovery of kinase inhibitors and new AhR binders are two real world applications of FROGS presented in this study. Some potential additional applications are also outlined on page 9:

“Thus, FROGS vectors are widely applicable to biomedical problems beyond drug target identification including identifying risk genes of diseases, uncovering disease-disease relationships, and single cell clustering and classification.”

The last sentence in the comment implies the reviewer might be concerned there are too many compounds predicted to target AhR. First, 76 out of 369 predictions were experimentally confirmed, suggesting the large number may be a desirable outcome instead of a shortcoming. Second, profiling 369 compound candidates does not pose an experimental burden in a modern drug discovery setting. Discoveries can still be made, even if logistical capability limits the experiment to test a few compounds. For example, 7 compounds could have been confirmed if the top 14 compounds were screened.

10. The manuscript requires significant language editing. There are several spelling and grammatical errors throughout. The authors should also refrain from using terms such as “more effective” or “high-quality” when referring to the compound-target predictions from FROGS.

Our native English speaking co-authors have carefully revised the manuscript. We also removed those subjective terms as pointed out by the reviewer.

Reviewer #2 (Remarks to the Author):

What are the noteworthy results?

Chen H. et al. applied the word2vec concept to embed biological functions (from GO and RNA-Seq experiments) in genes, which can be used to assess the similarity between two perturbations. The authors show that their embedding strategy, named FROGS, outperforms other embedding methods such as OPA2Vec (pre-trained on PubMed abstract) to identify similarity in gene expression between perturbations. The authors used FROGS to predict compound-target relationships, which they confirmed experimentally in two cases. The paper is well-written, and the results for compound-target prediction are of high interest. Whether FROGS could be used for other tasks like PPI prediction or gene-disease associations has not been shown in the paper, but it looks like other potential application of FROGS.

Will the work be of significance to the field and related fields? How does it compare to the established literature? If the work is not original, please provide relevant references.

The results are in line with the literature. When there were discrepancies, ex with CMap, they were highlighted in the supplementary notes. A section on the reasons why the authors think FROGS outperforms other embedding methods to assess the similarity of two perturbations (in Figure 1) should be added to the discussion of the paper to bonify the work.

The success of FROGS is likely due to the better capture of gene's functions provided by GO and ARCHS4, while other methods in Figure 1c either do not capture gene functions (e.g., Fisher's exact test and c-Map) or do not use GO and ARCHS4. This is discussed on page 8 and further detailed in Supplementary Note 1.

On page 8:

“A significant contribution of our study lies in the transfer of the word2vec concept from the NLP domain to bioinformatics problems, wherein genes are embedded into vectors representing diverse biological information, including their known functions in GO and empirical functions in ARCHS4. This has two significant implications. First, we validated that casting gene identities into their functional roles, via FROGS, resulted in weak pathway signals within large-scale data becoming much more readily extractable (Fig. 1c). The challenge of the weak overlap signals between two experimentally derived gene signatures is not due to the sparseness of the L1000 gene signatures, as we further tested Model *L* based on 1000 Landmark genes alone without seeing a qualitative difference with respect to the extrapolated whole-transcriptome signatures. FROGS's representation of a gene or a gene list as a numerical vector incorporating the genes' prior knowledge (Fig. 1b) is conceptually much more empowering compared to a one-hot encoding representation. Another explanation for the significantly better performance of FROGS (Fig 2.b) is that other representations do not model GO, embed GO indirectly via text annotations, or do not utilize a GO hypergraph (Supplementary Note 1).”

We added the following to Supplementary Note 1:

“While OPA2Vec takes into account GO annotations of genes which are treated as sentences, it does not explicitly model the co-functions of genes and the co-occurrence of GO terms. In comparison, this information can be reflected in our GO hypergraph. Gene2vec embeds genes solely based on gene expression data, overlooking the essential GO annotations that directly reflect gene functions. On the other hand, clusDCA embeds individual GO terms but does not consider each gene as a union of multiple functions. These differences might explain why FROGS, in its modeling of gene functions, demonstrates significantly improved performance.”

Does the work support the conclusions and claims, or is additional evidence needed?

The figures and supplementary tables do support the claims of the paper.

Are there any flaws in the data analysis, interpretation and conclusions? Do these prohibit publication or require revision?

I only have minor corrections:

page 2 paragraph 2: replace "a well-established statistical framework" with "well-established statistical frameworks"

Corrected.

page 3 paragraph 3: replace "the same color" by "same biological function(i.e. node color in Figure 1b)".

Corrected

page 4 paragraph 2: Did the authors consider comparing FROGS with rank-rank based similarity metrics such as RRHO (Plaisier SB et al, 2010 NAR) and if embedding methods would also outshine those type methods (they should perform better than Fisher's exact test)?

We thank the reviewer for the suggestion. RRHO was designed for genome-wide expression profile comparisons from two continuous ranked gene lists. In our simulation comparison discussed on page 4 paragraph 2, we simulated sets of genes without ranking. Therefore, RRHO cannot be directly applied to this data for comparison. In addition, RRHO identifies significant overlap between two ranked gene lists by determining the degree of statistical enrichment using the Fisher's exact test sliding across all possible thresholds through the two ranked lists. In our comparison, we compared with Fisher's exact test on gene sets without ranking and demonstrated that FROGS significantly outperforms Fisher's exact test. FROGS can be easily extended to compare ranked gene lists by using the same strategy of trying all possible thresholds. As FROGS outperforms Fisher's exact test, which is the fundamental component of RRHO, extending FROGS to ranked gene lists is anticipated to yield superior performance compared to RRHO.

page 4 paragraph 3: How many compounds had more than 5 targets and were excluded from the L1000 database?

We added to page 15:

“which included 2433 (c, g) pairs formed by 509 polypharmacological compounds (>5 targets/compound) and 507 targets.”

Is the methodology sound? Does the work meet the expected standards in your field?

The methodology is sound, and the method section (and supplementary notes) provide adequate details to understand the methodology.

Is there enough detail provided in the methods for the work to be reproduced?

The Python code to rerun some of the FROGS analysis is provided in GitHub but would benefit from better walkthrough examples.

We updated our GitHub repository. We have now provided an additional example with guidance on how users can use FROGS embeddings to build their own applications. Specifically, in this example, we show how FROGS embeddings can be used to help classify tissue-specific genes. We also included more detailed descriptions and usage instructions in the GitHub page.

Reviewer #3 (Remarks to the Author):

In current bioinformatics research, many machine learning applications rely on one-hot-encoding strategy (i.e. gene identities only), often missing the opportunity to leverage existing knowledge about gene functions. In response to this, the authors propose a novel approach that maps genes to a high dimensional numerical vector through the functional representation of genes, named FROGS. The method reimagines gene signatures, projecting them onto their biological functions rather than their identities, akin to the way the word2vec technique operates in natural language processing.

The authors validate the proposed gene embedding using different tests against a few existing gene embeddings,

including Gene2Vec, OPA2Vec-Gene and OPA2Vec-GO. First, FRoGS facilitates extraction of weak pathway signals from gene signature more significantly than the other gene embeddings.

Second, the authors experimental investigation showcases the remarkable potential of FRoGS when applied to L1000 datasets. It outperformed other models, resulting in significantly more accurate compound-target predictions. By seamlessly integrating additional pharmacological activity data sources, FRoGS makes a substantial contribution to generating high-quality compound-target predictions.

Third, the proposed gene embedding results in good prediction methods with a small number of parameters that hence require less training data than gene-identity-based machine learning methods.

Furthermore, the authors illustrated that FRoGS not only facilitated the prediction but also the recovery of chemical probes. Specifically, they found that when employing the combined model, 57% of network edges, 45% of kinase inhibitors, and 50% of AhR ligands would have been overlooked if relying solely on Model pQSAR.

Their validation studies highlight the broad applicability of FRoGS in the realm of machine learning-based bioinformatics. By equipping prediction networks with prior knowledge of gene functions, we can expedite the discovery of relationships among gene signatures obtained from large-scale OMICs studies encompassing compounds, cell types, disease models, and patient cohorts.

The content presented thus far is commendable. However, there is room for enhancement in the manuscript. Throughout the document, certain sentences appear awkward and ambiguous, while a few terms require clear definitions. Detailed comments addressing these issues can be found below.

In summary, this is a valuable research. It advances gene embedding techniques. However, it needs further revision before the acceptance decision is made.

Minor comments

1. Page 2.

In the first paragraph, provide the full name of shRNA which appears for the first time.

Corrected.

In the second paragraph, what does “human host” mean?

Changed to:

“influenza host dependency factors”

2. *In the first sentence of the results section “FRoGS vectors were pre-trained such that individual human genes ...”, The word “pre-trained” should be “pre-training”. Otherwise, how the model is pre-trained is not given. The author should describe how they pre-train the model in the Method section.*

We see how “pre-trained” can cause confusion and we change it to “trained”. The details of datasets and algorithms for training embeddings of individual genes encoding functional information were provided on page 10.

3. Page 5.

What are “orthogonal data sources”? The short explanation is necessary here, although such a term is discussed in the supplemental notes.

We modified the section title on page 5:

“FRoGS predicts compound targets supported by ~~orthogonal~~ **structure and activity** data sources”

Fig. 1b demonstrates the proposed method leads good correlation between genes with similar functions. The author states: ‘Genes closely positioned in the same clusters tend to share the same color ($p < 10^{-100}$)’. However, the visualization does not support this highly significance. Describe how to compute the p -value.

We added to page 12:

“Statistical analysis

Given a gene associated with a GO process G , we computed the mean μ and the standard error δ for its one-nearest neighbor (1-NN) to share the same GO process. Z score is defined as $(\mu - \mu_0)/\delta$, where μ_0 is the percentage of genes in the genome that associated with G . For example, among the 2476 genes associated with “immune system process”, the average fraction of their 1-NNs to be also associated with the same process is 0.57 ± 0.01 , which is higher than the probability that the “immune system process” occurs by chance ($\mu_0 = 0.14$) with a Z score of 43. All four GO processes examined in Fig.1b have p -values $< 1^{-100}$.”

Revise “The application of Model L to both training and test datasets resulted in the prediction of 780,438 compound-target pairs with a probability above 0.8.” It is confusing.

We improved this (page 5):

“The application of Model L to all compound-gene pairs, captured in the full L1000 dataset regardless of whether target annotations were available, resulted in the prediction of 780,438 compound-target pairs with probability values above 0.8.”

The statement “covering 1601, 93, and 321 Broad compounds, respectively” is not clear, with only 1438 drugs are used as the positive pairs in training set. ‘a Validation category for 2491 Model L predictions’ in Figure 3.a seems inconsistent with “2477 predictions fall into the three categories assigned based on their best supporting evidence.’

Model L was applied to compounds in the test set as well. Therefore, the compound counts noted included Broad compounds from both training set and test set. The corresponding number of compounds belong to the training set are 1060, 45, and 166, respectively.

We are grateful for the reviewer to find the discrepancy in the two numbers. Figure 3a was made with 2491 predictions (see table below, included as source data) and we replaced the number 2477 with 2491

(page 5). The number 2477 was obtained from a previous run and differed by 0.5% due to the randomness in the model L training process. The number 2491 came from the following table:

Category	Counts	Percentage
pQSAR	1756	70.5%
known	473	19.0%
pIC50	126	5.1%
structure sure	17	0.7%
structure likely	88	3.5%
NCI60	23	0.9%
PSP	8	0.3%
Total	2491	100%

4. Page 11. What are “noise” genes?

The term “noise genes” is used in the following context (page 12):

Due to stochasticity in biological signals and different choices of parameters in data preprocessing steps, “noise genes,” which are genes that do not directly associate with the pathways underlying the phenotype of interest may be included in the discovered gene set (non-pathway genes in Fig. 1a).

The meaning of “noise” is explained in the context as “do not directly associate with the pathways underlying the phenotype of interest”. The noise genes refer to non-pathway genes that ended up in the gene signatures. This could be either due to biological or technical variations in the dataset, or truly biologically relevant genes that are not explainable based on the current knowledge captured in Gene Ontology. For the purpose of data analysis, these genes diluted the interpretable biological signals and make the true biological pathways harder to detect, therefore, we refer to them as noise. We improved the sentence.

5. Page 12

Pairwise product $D(x_cpd) \odot D(x_gene)$ seems not meaningful. Given a brief explanation.

\odot denotes the element-wise product, which is a binary operation that takes in two vectors of the same dimensions and returns a vector of the multiplied corresponding elements. In this study, if the gene is the compound’s target, $D(x_cpd)$ should be similar to $D(x_gene)$ and their element-wise multiplication results in mostly positive elements in the output tensor. Otherwise, if the compound-gene pair is unrelated, the resultant tensor elements are less likely to be positive. Therefore, the resultant tensor can be fed into the downstream classification layers to generate the probability of the input gene being the target of the input compound. We hope this clarifies the biological reasoning behind this operation.

6. Page 14. “perturbagens” should be a typo.

The original context was:

“L1000 dataset has a rich experimental design comprising multiple cell lines and multiple genomic perturbagens reagents per gene g in the forms of both loss-of-function shRNA and gain-of-function cDNA treatments;”

We reparsed it (page 14):

“The expansive collection of gene signatures associated with the L1000 dataset is based on treatments using genomic reagents (both loss-of-function shRNA and gain-of-function cDNA treatments for each gene g) and compounds c dosed at multiple concentrations and durations screened across multiple cell lines.”

Reviewer #1 (Remarks to the Author):

The authors have addressed all the concerns raised during the first review. The revised version, especially the methods (with more details) and "Discussion" (more balanced) sections are much improved. I do not have any additional concerns.

Reviewer #2 (Remarks to the Author):

The authors answered all my questions/edits in this new manuscript version.

Reviewer #2 (Remarks on code availability):

The authors added an example of code to use FROGS to annotate tissue-specific genes, making the code more accessible.